# LET 2D DIFFUSION MODEL KNOW 3D-CONSISTENCY FOR ROBUST TEXT-TO-3D GENERATION

**Junyoung Seo**[*1]   **Wooseok Jang**[*1]   **Min-Seop Kwak**[*1]   **Hyeonsu Kim**[1]   **Jaehoon Ko**[1]
**Junho Kim**[2]   **Jin-Hwa Kim**[†2,3]   **Jiyoung Lee**[†2]   **Seungryong Kim**[†1]

[1]Korea University   [2]NAVER AI Lab   [3]AI Institute of Seoul National University

{se780,jws1997,mskwak01,hyeonsu0305,kjh9604}@korea.ac.kr,
{jhkim.ai,j1nhwa.kim,lee.j}@navercorp.com,
seungryong_kim@korea.ac.kr

## ABSTRACT

Text-to-3D generation has shown rapid progress in recent days with the advent of score distillation sampling (SDS), a methodology of using pretrained text-to-2D diffusion models to optimize a neural radiance field (NeRF) for a zero-shot setting. However, the lack of 3D awareness in the 2D diffusion model often destabilizes previous methods from generating a plausible 3D scene. To address this issue, we propose **3DFuse**, a novel framework that incorporates 3D awareness into the pretrained 2D diffusion model, enhancing the robustness and 3D consistency of score distillation-based methods. Specifically, we introduce a consistency injection module that constructs a 3D point cloud from the image generated from the text prompt and utilizes its projected depth map at a given view as a condition for the 2D diffusion model. The diffusion model, through its generative capability, robustly infers dense structure from the sparse point cloud depth map and generates a geometrically consistent and coherent 3D scene. We also introduce a new technique called semantic coding that reduces the semantic ambiguity of the text prompt for improved results. Our method can be easily adapted to various text-to-3D baselines. We experimentally demonstrate how our method notably enhances the 3D consistency of generated scenes compared to previous baselines, achieving state-of-the-art performance in geometric robustness and fidelity. The project page is available at https://ku-cvlab.github.io/3DFuse/

## 1   INTRODUCTION

Text-to-3D generation, the task of generating 3D content through given text prompts (Sanghi et al., 2022; Liu et al., 2022; Jain et al., 2022), has seen a rapid growth surge thanks to the advancements in diffusion models (Ho et al., 2020; Rombach et al., 2022). This innovation allows people to easily create 3D content without dealing with professional modeling tools, gaining significant attention across a wide range of industries, such as gaming, graphics, VR/AR, and animation.

One of the most popular methodologies currently used for text-to-3D generation tasks is score distillation sampling (SDS) (Poole et al., 2022), which employs the gradients of 2D diffusion models to optimize a 3D representation, *i.e.*, a Neural radiance field (NeRF) (Mildenhall et al., 2020). This approach has gained prominence due to its leveraging the generation capability of pretrained 2D diffusion models to create expressive and detailed 3D scenes. However, various works (Wang et al., 2023; Shi et al., 2023) show that 3D scenes generated by SDS often display distortions and artifacts depending on text prompts. One notable failure case is a 3D inconsistency problem (dubbed the "Janus problem" (Wang et al., 2023; Shi et al., 2023)), where a frontal feature (such as a frontal face) pertaining to the text prompt is replicated at other sides of the generated scene. As shown in (a) of Fig. 1, this problem arises from the inherent limitation of the 2D diffusion model – it lacks

---

[*]Equal contributions.
[†]Co-corresponding authors.

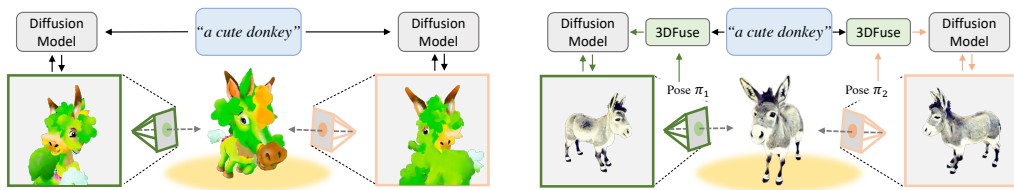

Figure 1: **Motivation.** (a) Previous methods (Poole et al., 2022; Song et al., 2020) solely use noised rendered images and text prompts for score distillation, often resulting in scenes with poor 3D consistency. Our **3DFuse** addresses this issue by giving diffusion models 3D awareness through a consistency injection module, drastically improving the 3D consistency of generated scenes.

awareness of the camera pose from which it views the scene. Therefore, it is prone to produce frontal geometric features across all scene directions, generating distorted and unrealistic 3D scenes.

A possible approach to solve this issue would be training a diffusion model that can be conditioned explicitly on camera pose values to produce plausible images of novel viewpoints (Liu et al., 2023). However, there are inherent hurdles to this approach. As the pose-conditioned diffusion model operates solely on the 2D domain and has no explicit knowledge of 3D space, it has difficulty in modeling 3D transformations as well as subsequent self-occlusions, resulting in geometrically inconsistent novel view inferences (Liu et al., 2024). The training also requires ground-truth 3D data, whose creations require either human effort or expensive 3D sensors, leading to smaller data size and inferior quality compared to that using the vast 2D data (Schuhmann et al., 2022) available. This limits the generative capability of this model from reaching that of ordinary 2D diffusion models. We solve this dilemma between 2D diffusion models and explicit 3D geometry with a middle-ground approach that combines the best of both worlds – a pretrained 2D diffusion model imbued with 3D awareness, conditioned on coarse yet explicit 3D geometry.

We propose a novel methodology, dubbed **3DFuse**, to improve the geometric consistency of generated 3D scenes. Central to our method is a consistency injection module, which effectively imbues pretrained 2D diffusion models with 3D awareness and is easily adaptable to existing SDS-based text-to-3D baselines. Specifically, our consistency injection module creates a 3D coarse geometry of given text through an off-the-shelf point cloud generation model (Nichol et al., 2022; Wu et al., 2023), which can be projected to arbitrary views to generate coarse depth maps. As this coarse depth map is explicitly 3D consistent and outlines the overall structure of the scene expected at the viewpoint, the diffusion model conditioned on it can optimize the given viewpoint in a 3D-aware manner. To this end, we add a ControlNet (Zhang & Agrawala, 2023) module to the diffusion model and fine-tune it, leveraging the generative capability of the pretrained diffusion model toward inferring dense structures of the scene despite the sparsity of the depth map. We also introduce semantic coding to add controllability and specificity in the generation process and show how it can be leveraged with Low-rank adaptation (LoRA) (Hu et al., 2021) for more semantically consistent results.

To demonstrate our method's adaptability and its universally powerful performance at various baselines, we employ our framework across a range of text-to-3D baseline models, *i.e.* Dreamfusion (Poole et al., 2022), SJC (Wang et al., 2022a), and ProlificDreamer (Wang et al., 2023). The results show significant improvements in both generation quality and geometric consistency on all baselines. We extensively demonstrate the effectiveness of our framework with qualitative analyses and ablation studies. Moreover, we introduce an innovative metric for quantitatively assessing the 3D consistency of the generated scenes.

## 2 RELATED WORK

**Diffusion models** (Ho et al., 2020; Song et al., 2021) have gained much attention as generative models due to their stability, diversity, and scalability. Given these advantages, diffusion models have been applied in various fields, such as image translation (Rombach et al., 2022; Seo et al., 2022) and conditional generation (Kanizo et al., 2013; Rombach et al., 2022). Especially, text-to-image generation has been highlighted with the introduction of various guidance techniques (Ho & Salimans, 2021; Dhariwal & Nichol, 2021). GLIDE (Nichol et al., 2021) utilizes CLIP (Radford

et al., 2021) guidance to enable text-to-image generation, followed by large-scale text-to-image models such as DALL-E2 (Ramesh et al., 2022) and Stable Diffusion (Rombach et al., 2022). Such emergence has led to the utilization of pretrained text-to-image models for tasks such as endowing additional conditions (Wang et al., 2022b) or performing manipulations (Gal et al., 2022; Kwon et al., 2022).

**Text-to-3D generation** models generally employ pretrained vision-and-language models, such as CLIP (Radford et al., 2021) to generate 3D shapes and scenes from text prompts. DreamFields (Jain et al., 2022) incorporates CLIP with neural radiance fields (NeRF) (Mildenhall et al., 2020), demonstrating the potential for zero-shot NeRF optimization using only CLIP as guidance. Recently, Dreamfusion (Poole et al., 2022) and SJC (Wang et al., 2022a) have demonstrated an impressive ability to generate NeRF with frozen diffusion models instead of CLIP. Concurrently, some approaches bring performance improvement through coarse-to-fine pipeline such as Magic3D (Lin et al., 2022) and Fantasia3D (Chen et al., 2023). ProlificDreamer (Wang et al., 2023) demonstrates extremely high-fidelity results through utilizing a variational version of SDS, named VSD. However, despite their impressive performance, the 3D inconsistency problem fundamentally remains, causing them to often generate distorted geometries.

**Image-to-3D generation** models generate 3D scenes from a conditioning image, which is analogous to the task of generation-based novel view synthesis. There are 3D native diffusion models such as Point-E (Nichol et al., 2022), trained upon several million internal 3D models to generate point clouds, as well as SDS-based methods such as NeRDi (Deng et al., 2023). NeuralLift-360 (Xu et al., 2023) and Realfusion (Melas-Kyriazi et al., 2023) utilize reconstruction loss combined with monodepth estimation and textual inversion, respectively, to strengthen the semantic alignment of 3D scene to the image. Among these, a concurrent work Zero123 (Liu et al., 2023) bears similarity to our work as it models 3D scenes in the 2D image domain by finetuning a 2D diffusion model to infer a novel view of the input image given relative camera pose. Our work diverges from this approach in that we model the coarse 3D geometry explicitly. This explicit geometry enables a view-aligned depth map to be given as a condition for the diffusion model, allowing it to model 3D consistency more effectively.

## 3 PRELIMINARY

For the task of text-to-image generation, diffusion models such as Stable Diffusion (Rombach et al., 2022) receive a text prompt as an additional condition. Specifically, when a text prompt $c$ is given, a mapping model $\mathcal{T}(\cdot)$ maps the prompt $c$ into the embedding $e = \mathcal{T}(c)$. Then, the embedding $e$ is injected into the diffusion model with parameters $\theta$. Formally, we denote a score prediction network of the text-to-image diffusion model as $\epsilon_\theta(x_t, t, \mathcal{T}(c))$, where $x_t$ represents a noisy image with a noise level $t$ added to a clean image $x_0$. For the sake of brevity, we shall omit the variable $t$ and refer to the function $\epsilon_\theta(x_t, \mathcal{T}(c))$.

Score distillation sampling (SDS) (Poole et al., 2022) optimizes NeRF parameters toward following the direction of the score predicted by a frozen diffusion model towards higher-density regions. Specifically, let us denote $\Theta$ as parameters of NeRF, and $\mathcal{R}_\Theta(\pi)$ as a rendering function given a camera pose $\pi$. In SDS, a random camera pose $\pi$ is sampled, and the diffusion model is utilized to infer the 2D score of the rendered image, *i.e.*, $x = \mathcal{R}_\Theta(\pi)$. This score is used to optimize the NeRF parameters $\Theta$ by letting the rendered image move to the high-density regions, *i.e.*, to be realistic. It can be understood as minimizing a standard noise prediction loss function (Ho et al., 2020) with respect to the NeRF parameters $\Theta$ instead of the diffusion model's parameters $\theta$ such that:

$$\Theta^* = \underset{\Theta}{\operatorname{argmin}} \, \mathbb{E}_{t,\epsilon,\pi}\Big[w(t)\big\|\epsilon_\theta\big(x_t, \mathcal{T}(c)\big|x = \mathcal{R}_\Theta(\pi)\big) - \epsilon\big\|_2^2\Big], \tag{1}$$

where $\epsilon$ is a Gaussian noise and $w(t)$ is a weighting function. The Jacobian term of diffusion U-net $\partial\epsilon_\theta\big(x_t, \mathcal{T}(c)\big)/\partial x_t$ from the gradient of Eq. 1 can be omitted, formulating the new gradient as:

$$\nabla_\Theta \mathcal{L}_{\text{SDS}}(x = \mathcal{R}_\Theta(\pi); \theta) \triangleq \mathbb{E}_{t,\epsilon}\Big[\tilde{w}(t)\big(\epsilon_\theta\big(x_t, \mathcal{T}(c)\big) - \epsilon\big)\frac{\partial x}{\partial\Theta}\Big], \tag{2}$$

$\tilde{w}(t)$ is a weighting function from the DDPM (Ho et al., 2020) diffusion process.

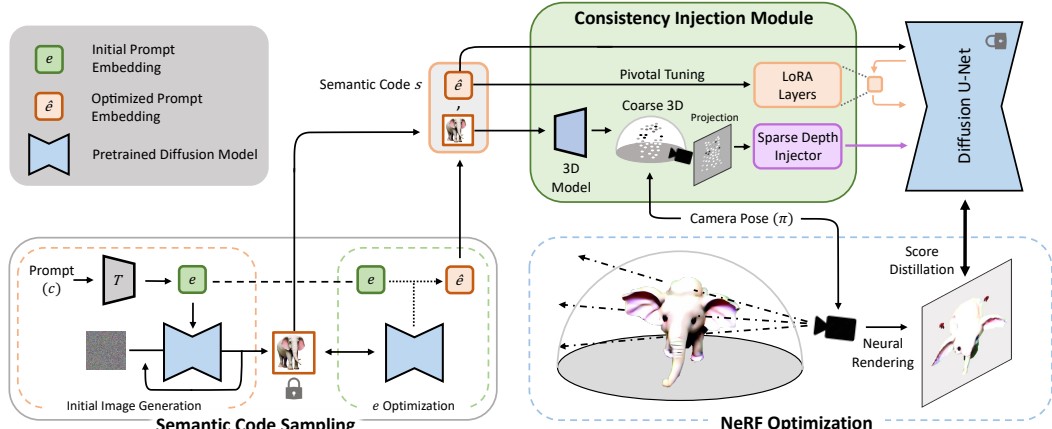

Figure 2: **Overall architecture of 3DFuse.** Our framework consists of semantic code sampling, followed by a consistency injection module that gives 3D-aware conditions to the diffusion model through a sparse depth injector. Semantic code, along with LoRA (Hu et al., 2021) layers, helps maintain the semantic consistency of 3D generation.

## 4 METHOD

### 4.1 MOTIVATION AND OVERVIEW

The method of score distillation-based text-to-3D generation (Poole et al., 2022), despite its effectiveness, currently possesses a crucial problem: inconsistent and distorted geometry in generated scenes due to 2D diffusion models not having explicit awareness of the 3D space nor the camera viewpoint it is optimizing from (Wang et al., 2023). Previous works (Poole et al., 2022; Wang et al., 2022a) attempt to circumvent this problem by adding the text prompts that roughly describe the camera viewpoint (*e.g.*, "*side view*"). However, this ad-hoc approach is severely limited, as a wide range of different pose values is ambiguously represented by few text prompts, leaving the generation process still vulnerable to geometric inconsistencies and severe deformations. Therefore, to overcome this problem, the objective boils down to how we can inject more specific and explicit 3D awareness into pretrained 2D diffusion models.

Our method, **3DFuse**, achieves this objective by conditioning a 2D diffusion model to projected depth maps of coarsely generated 3D structures. Specifically, we first conduct semantic code sampling, in which we initially generate an image $\hat{x}$ from the given text prompt to specify the semantics of the 3D scene we wish to generate (Sec. 4.2). From this image $\hat{x}$, we construct a coarse point cloud using an off-the-shelf point cloud generation model (Nichol et al., 2022; Wu et al., 2023). For every rendering view, we leverage its projected depth maps as explicitly 3D consistent conditions for the 2D diffusion model. As the sparse depth map contains rich 3D information describing the scene from a given viewpoint, this approach effectively enables the diffusion model to generate the 3D scene in a 3D-aware manner (Sec. 4.3). We further leverage the semantic code through low-rank adaptation (LoRA) (Hu et al., 2021) layers to ensure semantic consistency of the scene (Sec. 4.4). The overall architecture of **3DFuse** is described in Fig. 2.

### 4.2 SEMANTIC CODE SAMPLING

We focus on the fact that when generating a 3D scene from a text, inherent ambiguity exists within the text prompt. For instance, the text prompt "*a cute cat*" is ambiguous regarding color, as it could refer to either a black or a white cat. Such lack of specificity leaves the possibility for the 3D scene to be erroneously guided toward vastly different textures and semantics at different viewpoints, resulting in a semantically inconsistent 3D scene overall, as further demonstrated in Sec. 5.4.

We introduce a simple yet effective technique to reduce such text prompt ambiguity called semantic code sampling. To specify and secure the semantic identity of the scene we want to optimize towards, a 2D image $\hat{x}$ is generated from the text prompt $c$. Then, we optimize the text prompt embedding $e$ to better fit the generated image, similarly to the textual inversion (Gal et al., 2022):

$$\hat{e} = \underset{e}{\arg\min} \, ||\epsilon_\theta(\hat{x}_t, e) - \epsilon||_2^2, \tag{3}$$

where $\hat{x}_t$ is a noised image of the generated image $\hat{x}$ with the noise $\epsilon$ and the noise level $t$. We refer to this pair of the generated image $\hat{x}$ and the optimized embedding $\hat{e}$ as a semantic code $s$, *i.e.*, $s := (\hat{x}, \hat{e})$, which would be an input for our consistency injection module.

### 4.3 INCORPORATING A COARSE 3D PRIOR

Our main objective is injecting 3D awareness into a pretrained diffusion model to enhance the 3D consistency of generated scenes, while fully leveraging the 2D diffusion model's expressive generative capabilities. We achieve this with a novel consistency injection module, which conditions diffusion models on sparse depth projections of a constructed point cloud.

Specifically, we employ an off-the-shelf 3D model $\mathcal{D}(\cdot)$ to construct a 3D point cloud from the initial image $\hat{x}$ included in the semantic code $s$. $\mathcal{D}(\cdot)$ can be chosen from a wide variety of models: it could be a point cloud generative model such as Point-E (Nichol et al., 2022) or a single-image 3D reconstruction model such as MCC (Wu et al., 2023). For every SDS optimization step, as an image of the 3D scene is rendered using the diffusion model at camera viewpoint $\pi$, the constructed point cloud is projected to the same viewpoint $\pi$ resulting in a point cloud depth map $p$:

$$p = \mathcal{P}(\mathcal{D}(\hat{x}), \pi), \tag{4}$$

where $\mathcal{P}(\cdot, \pi)$ is a depth-projection function with a camera pose $\pi$. Adopting the architecture of ControlNet (Zhang & Agrawala, 2023), our *sparse depth injector* $\mathcal{E}_\phi$ receives the sparse depth map $p$, and we add its output features to the intermediate features within pretrained diffusion U-net of $\epsilon_\theta(\hat{x}_t, \hat{e})$ in a residual manner, which can be further formulated as $\epsilon_\theta(\hat{x}_t, \hat{e}, \mathcal{E}_\phi(p))$.

The effectiveness of this approach can be understood from multiple aspects: most importantly, the sparse depth map $p$, explicitly modeled in 3D, provides the 2D diffusion model with the 3D consistent outline of the scene it is expected to generate, effectively encouraging 3D geometric consistency in the generation process. This approach also adds much-needed *controllability* to the text-to-3D generation pipeline: because our method enables the overall shape of the scene to be decided before the lengthy SDS optimization process, the user can pick and choose from a variety of initial point cloud shapes, allowing one to more easily generate specific 3D scenes tailored to their needs.

**Training the sparse depth injector.** As the sparse point cloud obtained by the off-the-shelf 3D model inevitably contains errors and artifacts, its depth map also displays artifacts, shown in Fig. 3(a). Therefore, our module must be able to handle the inherent sparsity and the errors present in the projected depth map.

To this end, we employ two training strategies for our sparse depth injector $\mathcal{E}_\phi(\cdot)$. First, we train our sparse depth injector using a paired dataset of real-life images and their point cloud depths (Reizenstein et al., 2021). Our model is trained to generate real-life images given its point cloud depth as a condition, enabling it to infer dense structures from sparse geometry given in point cloud depths. To increase the robustness of our model against errors and artifacts, we impose augmentations by randomly subsampling and adding noisy points to the point clouds from which the depth maps originate.

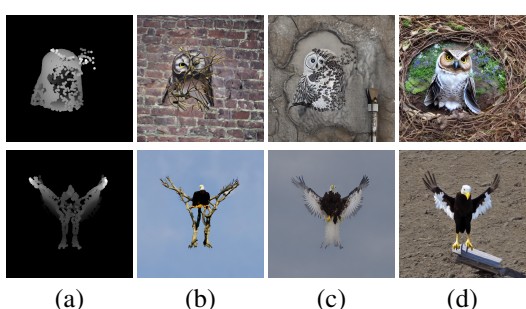

Figure 3: **Qualitative results conditioned on the sparse depth map.** Given sparse depth maps (a), (b) are results of depth-conditional Stable Diffusion, (c) are results of ControlNet trained on MiDaS depths only, and (d) are **3DFuse** results. Given text prompts are "*a front view of an owl*" and "*a majestic eagle*," respectively.

Second, the injector $\mathcal{E}_\phi(\cdot)$ is also trained on dense depth maps of text-to-image pairs, predicted using MiDaS (Ranftl et al., 2020). This strengthens the model's generalization capability, enabling it to infer dense structural information from categories not included in the 3D point cloud dataset for sparse depth training. In combination, given the depth map $p$ along with the corresponding image $y$ and caption $c$, the training objective of the depth injector $\mathcal{E}_\phi(\cdot)$ is as follows:

$$\mathcal{L}_{\text{inject}}(\phi) = \mathbb{E}_{y,c,p,t,\epsilon}\left[ ||\epsilon_\theta(y_t, c, \mathcal{E}_\phi(p)) - \epsilon||_2^2 \right]. \tag{5}$$

Note that only the depth injector $\mathcal{E}_\phi(\cdot)$ is trained while the diffusion model remains frozen, making the training process more efficient, akin to finetuning.

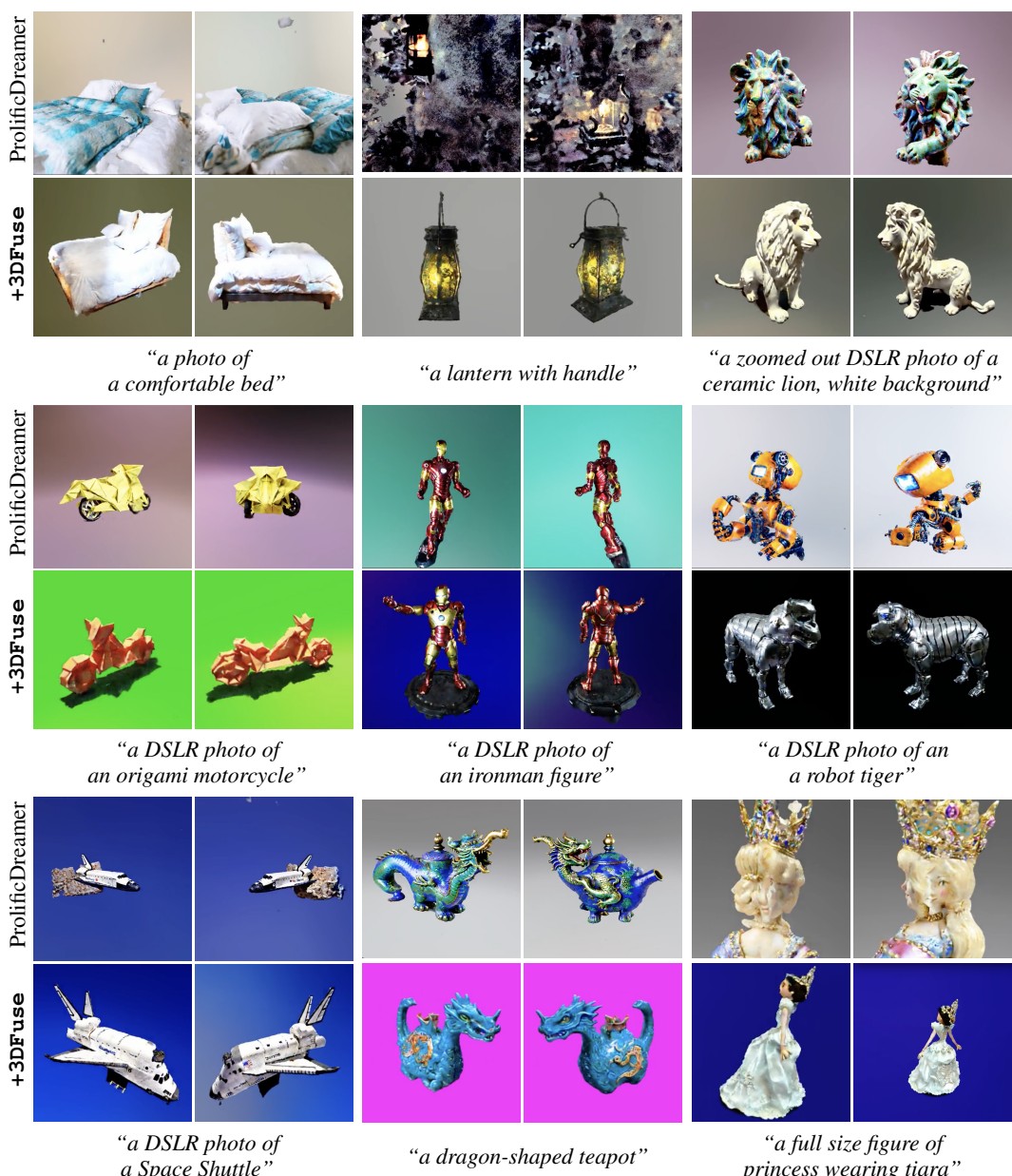

Figure 4: **Qualitative comparisons.** We demonstrate the effectiveness of our approach on a Prolific-Dreamer (Wang et al., 2023) baseline. Incorporation of **3DFuse** framework drastically enhances 3D consistency and fidelity of generated scenes.

These training strategies enable our model to receive sparse and noisy depth maps directly as input and successfully infer dense and robust structural information without needing an auxiliary network for depth completion. As shown in Fig. 3, our approach generates realistic results without being restricted to the domain of the point cloud dataset. Note the category of images (birds) used for the experiment is not included in the category of the point cloud dataset we use.

## 4.4 IMPROVING SEMANTIC CONSISTENCY

To result in a scene semantically consistent at all viewpoints, the diffusion model should produce a score that maintains a certain degree of semantic consistency regardless of given viewpoints. Although optimized embedding $\hat{e}$ partially achieves this, we further enhance this effect by adopting LoRA (Hu et al., 2021) technique motivated by (Ryu, 2022). LoRA layers with parameters $\psi$ consist of linear layers inserted into the residual path of the attention layers in the diffusion U-net.

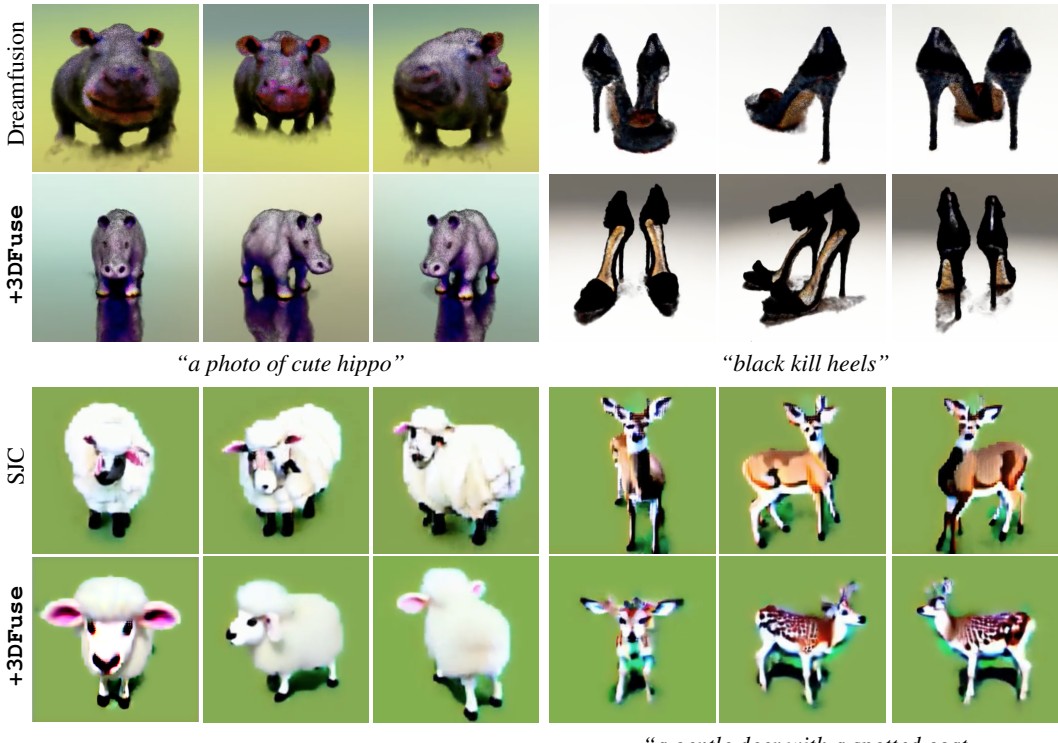

Figure 5: **3DFuse adaptation to other baselines.** We demonstrate the effectiveness our approach on Stable-DreamFusion (Poole et al., 2022) and SJC (Wang et al., 2022a) baselines. Incorporation of **3DFuse** drastically enhances 3D consistency and fidelity of generated scenes.

Training the LoRA layers in this manner instead of the entire diffusion model further helps it avoid overfitting to a specific viewpoint. In practice, given an image $\hat{x}$ generated from text prompt $c$, we fix the optimized embedding $\hat{e}$ and tune the LoRA layers $\psi$ (Roich et al., 2022):

$$\mathcal{L}_{\mathrm{LoRA}}(\psi) = \mathbb{E}_{\epsilon,t}\left[||\epsilon_{\theta,\psi}(\hat{x}_t, \hat{e}, \mathcal{E}_\phi(p)) - \epsilon||_2^2\right]. \tag{6}$$

## 5 EXPERIMENTS

### 5.1 IMPLEMENTATION DETAILS

We conduct all experiments with a Stable Diffusion model based on LDM (Rombach et al., 2022), and train the sparse depth injector using a subset of the Co3D dataset from the weights (Zhang & Agrawala, 2023) trained using the image-text pairs along with the depth maps predicted by MiDaS. We conduct experiments with two off-the-shelf modules, mainly using the Point-E and in certain experiments also leveraging MCC (Wu et al., 2023) with MiDaS. For the semantic code sampling, we adopt Karlo (Donghoon Lee et al., 2022) based on unCLIP (Ramesh et al., 2022), as we find that it tends to follow user prompt more closely.

### 5.2 TEXT-TO-3D GENERATION

We apply **3DFuse**'s methodology to previous SDS-based text-to-3D frameworks DreamFusion, Score Jacobian Chaining (SJC), and ProlificDreamer. All experiments utilize the Stable Diffusion as its diffusion model, which is a publicly available large-scale text-to-image diffusion model. Because DreamFusion's official implementation uses publicly unavailable Imagen (Saharia et al., 2022), we instead resort to using Stable Diffusion for implementation.

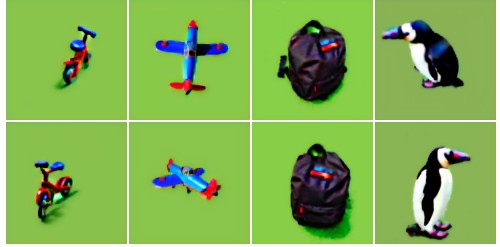

Figure 6: **Qualitative results of 3DFuse with MCC (Wu et al., 2023)**. Our **3DFuse** framework yields high-fidelity results with MCC model.

Figure 7: **Quantitative evaluation.** We compare 3D consistency with our proposed metric using the COLMAP Schonberger & Frahm (2016).

| Method | Variance ↓ |
|---|---|
| SJC + **3DFuse** (Ours) | **0.0499** |
| SJC Wang et al. (2022a) | 0.0870 |

Figure 8: **User study.** The user study is conducted by surveying 102 participants to evaluate 3D coherence, prompt adherence, and rendering quality.

| Method | 3D coherence | Prompt adherence | Overall quality |
|---|---|---|---|
| SJC + **3DFuse** (Ours) | **60.1%** | **59.2%** | **60.9%** |
| Stable-DreamFusion | 23.4% | 23.4% | 22.7% |
| SJC | 16.5% | 17.4% | 16.4% |

**Qualitative evaluation.** We present our qualitative evaluation in Fig. 4 and Fig. 5, which clearly demonstrates the effectiveness of our method in ensuring the geometric consistency and high fidelity of generated 3D scenes. While the baseline methods produce inconsistent, distorted geometry in multiple directions, combined with **3DFuse**, they are empowered to generate robust geometry at every viewpoint, as evidenced by the figure as well as video results provided in the supplementary materials. In Fig. 6, we also present the qualitative results of our approach when utilizing MCC (Wu et al., 2023) for the 3D prior, which also show consistent high-fidelity 3D scene generation results.

**Quantitative evaluation.** It is difficult to conduct a quantitative evaluation on a zero-shot text-to-3D generative model due to the absence of ground truth 3D scenes corresponding to the text prompts. In this light, we propose a new metric that utilizes COLMAP (Schonberger & Frahm, 2016) to measure the 3D consistency of a generated scene. In Table 7, we report the average variance scores over 42 generated 3D scenes. Our **3DFuse** framework outperforms baseline SJC by a large margin, demonstrating that our framework achieves more geometrically consistent text-to-3D generation. The details of our metric are given in Appendix C.2.

**User study.** We have conducted a user study with 102 participants, which is shown in Table. 8. We have asked the participants to choose their preferred result regarding 3D coherence, prompt adherence, and overall quality between Stable-DreamFusion, SJC, and our **3DFuse**-combined SJC. The results show that **3DFuse** generates 3D scenes that the majority of people judge as having higher fidelity and better geometric consistency than previous methods. Further details are described in Appendix C.1.

### 5.3 VIEW-DEPENDENT TEXT-TO-IMAGE GENERATION

We conduct text-to-image generation experiments with **3DFuse** to verify our framework's capability to inject 3D awareness into 2D diffusion models. We observe whether the images are generated in a 3D-aware manner when viewpoints are given as conditions through our **3DFuse** framework. Fig. 9 demonstrates the effectiveness of our framework: it allows for highly precise control of the camera pose in 2D images, with even small changes in viewpoint being reflected well on the generated images. Our approach shows superior performance to previous prompt-based methods (*e.g.*, "*A front view of*") regarding both precision and controllability of injected 3D awareness.

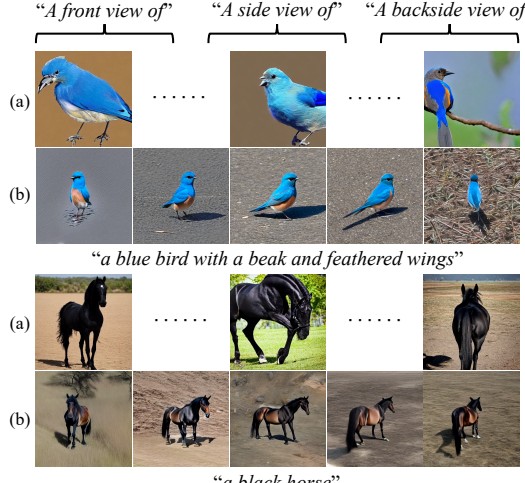

Figure 9: **View-dependent image generation comparison.** (a) are the results of naive view augmented prompting, and (b) our results with **3DFuse** framework.

### 5.4 ABLATION STUDY

**Comparison with Zero-123.** We conduct a qualitative comparison with concurrent work Zero-123, which bears similarity to our work in that it aims to inject 3D awareness into diffusion models through view conditioning. We compare the two methods by conducting SDS-based 3D scene generation, one with Zero-123 as its diffusion model and the other with our **3DFuse** framework as its diffusion backbone model, with all other settings identical. As shown in Fig. 10, we notice that our **3DFuse** results are generally more

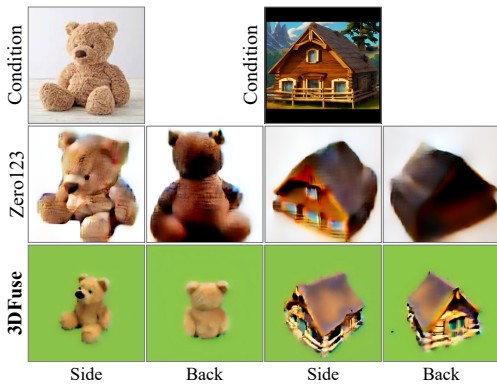

Figure 10: **Qualitative result for comparison with Zero-1-to-3 (Liu et al., 2023).** With reference image directly given as input, **3DFuse** generates more expressive, high-fidelity 3D scenes than Zero-1-to-3.

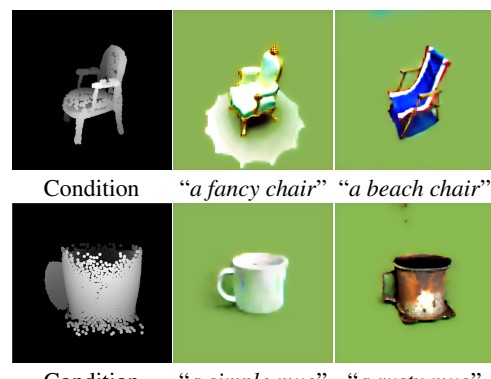

Figure 11: **3D reconstruction with different prompts given a single 3D structure.** The first column represents the given point clouds as conditions, and the second and third columns show synthesized results.

expressive and high-fidelity results. We hypothesize this phenomenon is related to the fact that, unlike Zero-123, which requires finetuning of the diffusion model itself for incorporation of 3D pose awareness, our framework only finetunes an external conditioning module in a manner similar to ControlNet (Zhang & Agrawala, 2023), leaving the diffusion model untouched. This allows our framework to leverage the pretrained diffusion model's generative capability to the fullest, resulting in more expressive and high-fidelity scene generation. We provide additional experiment results applying Zero-123 for text-to-3D generation, like our methodology, in section B of our Appendix.

**Different prompts with single 3D representation.** 3D representation, *i.e.* point cloud obtained from off-the-shelf models (Nichol et al., 2022; Wu et al., 2023) is typically coarse and sparse. To investigate how the diffusion model implicitly refines the coarse 3D structure, we conduct an ablation study using a fixed point cloud and different text prompts instead of inferring a point cloud from the initial image $\hat{x}$ of semantic code $s$. Fig. 11 shows that our **3DFuse** is flexible in responding to the error and sparsity inherent in the point cloud, depending on the semantics of text input.

**Semantic code sampling.** We conduct an ablation study on semantic code sampling in our **3DFuse** framework. The results in the first two columns of Fig. 12 show significant differences in geometry (pumpkin's smooth left surface and bumpy right surface) and semantic details (the tank's absence of wheel on the left) without semantic code sampling, depending on viewpoints. In contrast, using semantic code sampling ensures both geometric and semantic consistency across viewpoints, demonstrating its prominence in preserving the semantic identity of the 3D scene.

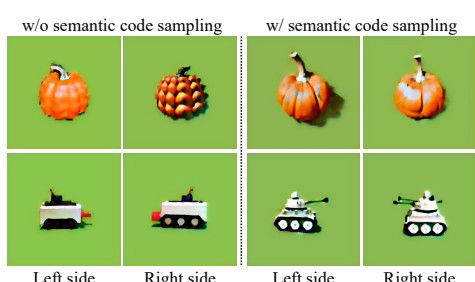

Figure 12: **Ablation study on semantic code sampling.** The given prompts are "*a round orange pumpkin with a stem and a textured surface*" (top) and "*a product photo of a toy tank*" (bottom).

## 6 CONCLUSION

In this paper, we address the 3D inconsistency problem within SDS-based text-to-3D generation. We propose a novel framework, **3DFuse**, that incorporates 3D awareness into a pretrained 2D diffusion model. Our method utilizes viewpoint-specific depth maps from a coarse 3D structure passed through a sparse depth injector and semantic code sampling for semantic consistency. Our work offers a practical solution for addressing the limitations of current text-to-3D generation techniques and opens up possibilities for more realistic 3D scenes from text prompts. Our experimental results demonstrate the effectiveness of our framework, outperforming previous models in quantitative metrics and qualitative human evaluation.

ACKNOWLEDGEMENTS

This research was supported by the MSIT, Korea (IITP-2024-2020-0-01819, ICT Creative Consilience Program, RS-2023-00227592, Development of 3D Object Identification Technology Robust to Viewpoint Changes).

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

# Appendix

In Sec. A, we provide additional implementation details for our proposed method, 3DFuse. In Sec. B, we present the results of additional experiments to validate our approach. In Sec. C, we provide details regarding the user study and the evaluation of our method. In Sec. D, we discuss the limitations of our approach.

## A  IMPLEMENTATION DETAILS

### A.1  TRAINING DETAILS

We use the Co3D (Reizenstein et al., 2021) dataset to train our sparse depth injector. The dataset is comprised of 50 categories and 5,625 annotated point cloud videos. From these point cloud-annotated videos, we randomly subsample three frames to create 16,875 pairs of RGB images and their projected depth map. We use 3k-5k points, augmented with 0-10% noise points, sampled from the dense point clouds provided in the Co3D dataset. We employ the rasterizer from Py-Torch3D (Ravi et al., 2020) library to project the point clouds and synthesize sparse depth maps. To relieve domain constraints from the Co3D dataset, we also employ text-to-image pairs along with MiDaS (Ranftl et al., 2020)-predicted dense depth maps. For ease of training, we start from the weights (Zhang & Agrawala, 2023) trained on the text-to-image pairs with MiDaS depth and fine-tune the model using the sparse depth maps synthesized from the Co3D dataset for 2 additional epochs.

### A.2  INTENSITY CONTROLS

All networks equipped with a pretrained diffusion model in our framework are designed to add external features to the intermediate features of the diffusion U-Net in a residual manner. Therefore, their intensities can be adjusted by multiplying scaling factors. This enables us to control the influence of the LoRA layers, used for semantic consistency, and the sparse depth injector, used for geometric consistency, on the diffusion model. In particular, adjusting the intensities of the LoRA layers is effective in controlling how much our 3D scene overfits the initial image. We use scaling factors of 0.3 and 1.0 on the features passing through the LoRA layers and the sparse depth injector, respectively.

### A.3  SEMANTIC CODE SAMPLING

To perform semantic code sampling, an initial image is first generated based on the text prompt, followed by optimization of corresponding text prompt embedding. The specific method follows that of Textual Inversion (Gal et al., 2022), wherein a new word is added to the embedding space and then optimized. Additionally, this initially generated image is used as input for the off-the-shelf 3D model in the auxiliary module. To facilitate the optimization and 3D shape inference processes, we add "*a front view of*" and "*, white background*" before and after the text prompt.

### A.4  ARCHITECTURAL CHOICES

There are manifold text-to-image diffusion models that have been trained at a large scale, including DALL-E2 (Ramesh et al., 2022), Imagen (Saharia et al., 2022), and Stable Diffusion (Rombach et al., 2022). However, as DALL-E2 and Imagen are not publicly available, we adopt Stable Diffusion, the most popular and widely used model.

To impose an additional condition on pretrained text-to-image diffusion models, various methods such as PITI (Wang et al., 2022b) and ControlNet (Zhang & Agrawala, 2023) have been proposed. Recently, Depth-conditional Stable Diffusion (Rombach et al., 2022) has been introduced, which finetunes Stable Diffusion to receive MiDaS (Ranftl et al., 2020) depth map as a condition for image translation. After conducting experiments with the above methods to find the optimal solution for our setting, we find that ControlNet achieves the best efficiency in training a sparse depth-conditioned diffusion model. We thus incorporate ControlNet architecture into our framework. Fig. 13 illustrates our architectural choice incorporating ControlNet architecture into our 3DFuse framework.

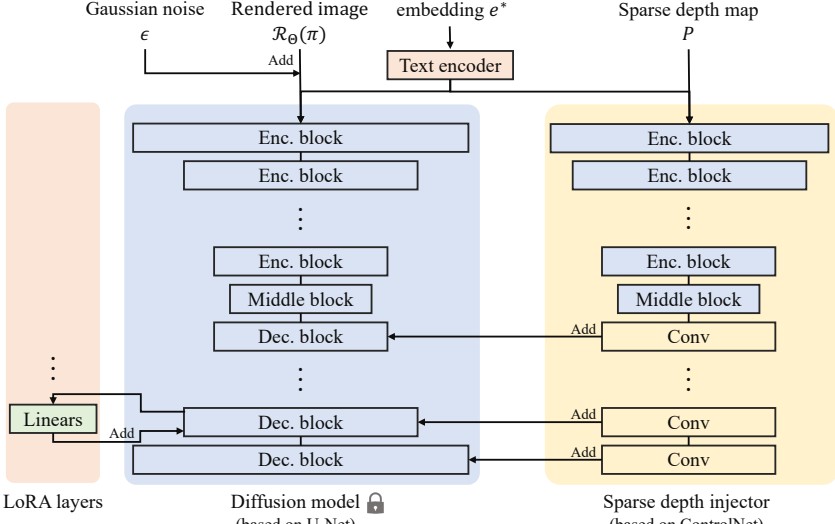

Figure 13: **Architectural choices for 3DFuse framework.** The sparse depth injector follows the architecture of ControlNet (Zhang & Agrawala, 2023), with the encoder side comprised of encoder blocks copied from the diffusion U-Net and the decoder side comprised of convolutional layers. The sparse depth map features are added to the intermediate features in the decoder blocks of the diffusion U-Net. LoRA layers are attached to every attention layer of the diffusion U-Net.

## B ADDITIONAL EXPERIMENTS

### B.1 COMPARISONS WITH NOVEL VIEW MODEL-BASED CONCURRENT WORKS

In Figure 14, we conduct a comparison of our model with different image-to-3D models (Liu et al., 2023; Qian et al., 2023), we first generate images from a single text prompt (through the Stable Diffusion (Metzer et al., 2022) model) and give these images Zero-123 (Liu et al., 2023) and Magic-123 Qian et al. (2023) for 3D scene generation. This experiment setting can be seen as observing what happens if the image-to-3D component of our 3DFuse methodology is replaced with other existing explicit image-to-3D models. Our 3DFuse undergoes its original generation process with the identical text prompt, and the images given to each model are not curated. Also, to remove the aliasing-like effect in the renderings that occurred due to the feature rendering architecture of SJC, we have changed the backbone NeRF model of Zero-123 and 3DFuse from feature-level SJC to pixel-level Instant-NGP backbone, which allows us to gain clearer results.

The experiment results show that our method's generative capabilities enable our model to generate more detailed, high-fidelity results in comparison to image-to-3D models that attempt explicit novel view synthesis of conditioning images. We hypothesize the reason for this is that strict image-to-3D models struggle to predict novel views of an image when an image is outside the domain of training data, while our model is loosely conditioned and therefore gives the generative model more freedom to generate realistic details, fully realizing its generative capability. This agrees with what we have explained in our comparison Zero-123 in our main paper.

### B.2 ROBUSTNESS

In the main paper, we present experimental results of 3DFuse with fixed random seeds for a fair comparison. Here, we conduct an additional experiment to verify the robustness of our approach. We compare our method with previous works (Poole et al., 2022; Wang et al., 2022a) with combinations of fixed text prompts and varying random seeds. Fig. 15 demonstrates that our approach exhibits enhanced robustness compared to previous works in terms of stochasticity.

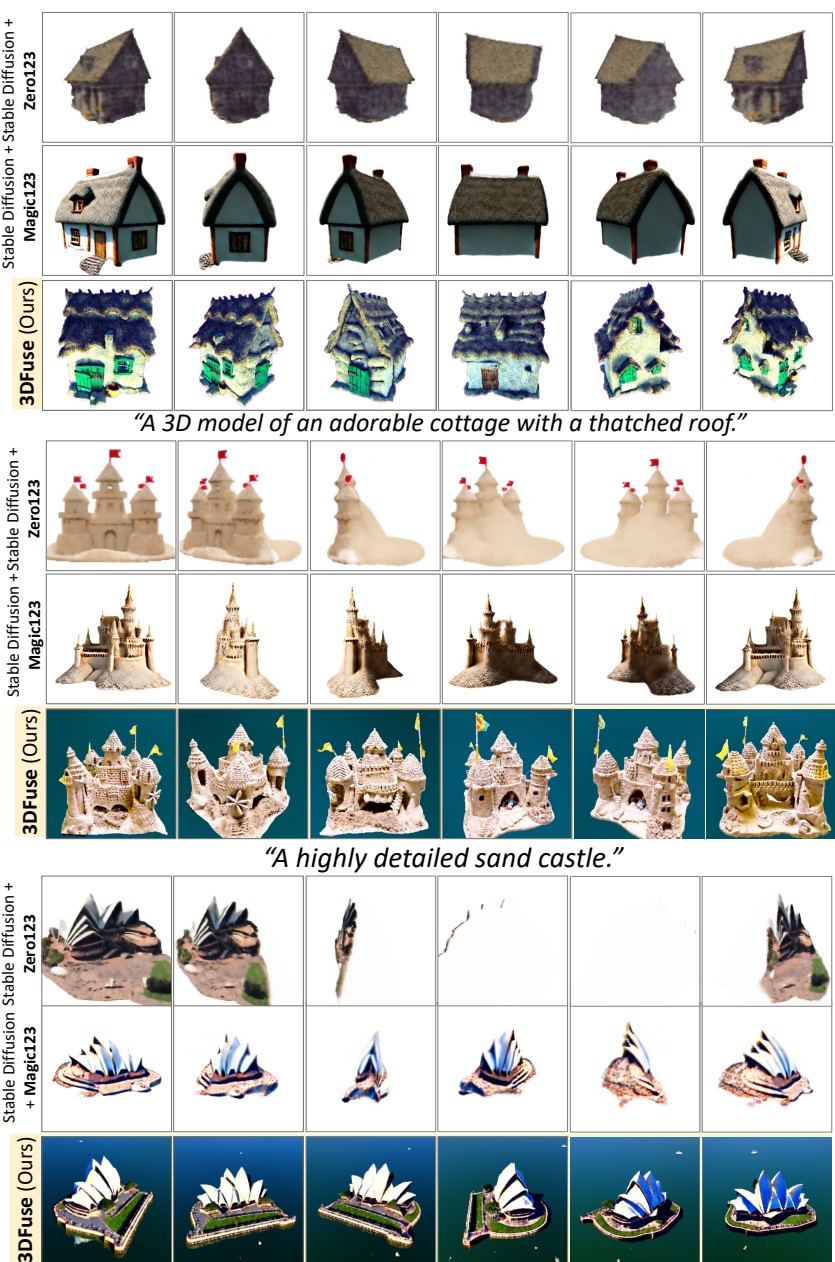

Figure 14: **Comparisons with novel view model-based concurrent works (Liu et al., 2023; Qian et al., 2023).** The experiment results show that our method's generative capabilities enable our model to generate more detailed, high-fidelity results in comparison to image-to-3D models that attempt explicit novel view synthesis of conditioning images.

### B.3 SPARSITY VARIATION

We provide additional analysis of our model's robustness against sparsity and ambiguity of point cloud. We drastically change the number of points in a point cloud and generate images from its depth map. As shown in Fig. 16, despite varying sparsity, our model shows consistency in inferring dense structures and generates realistic images.

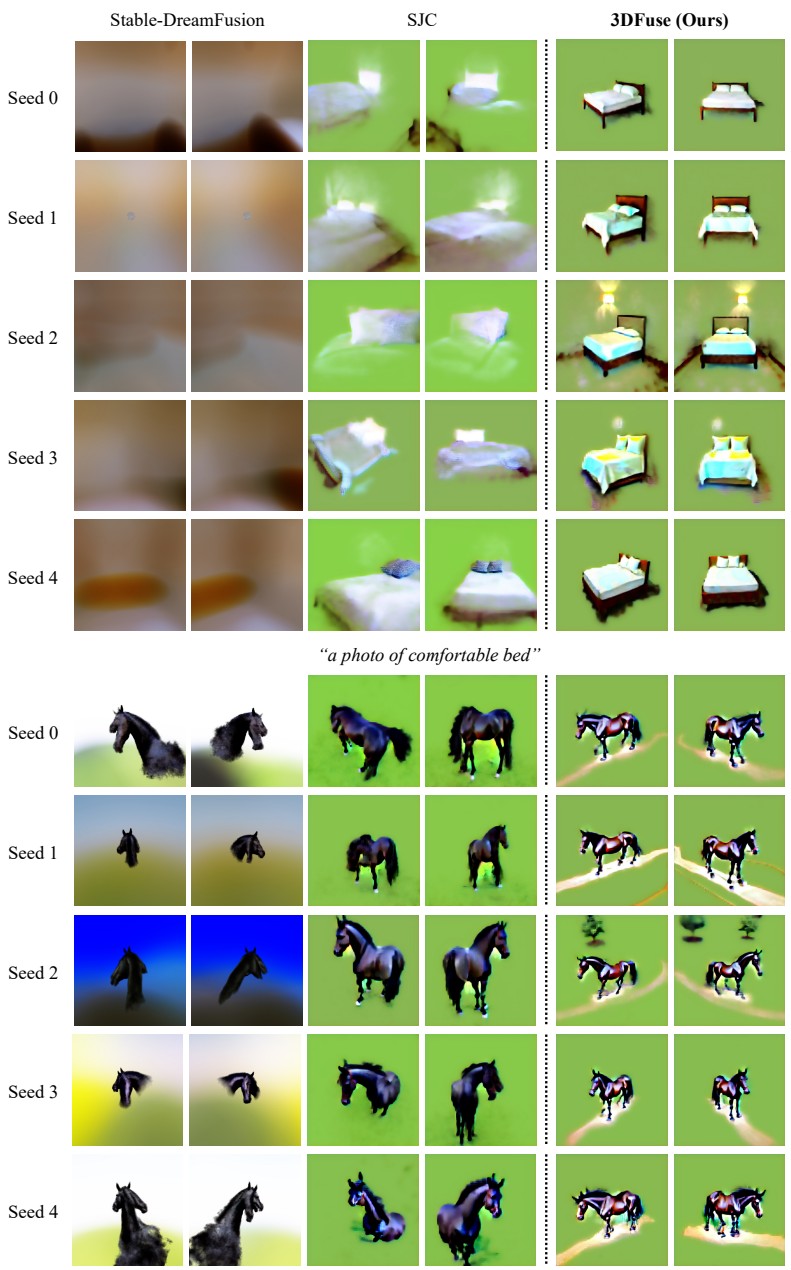

Figure 15: **Qualitative comparison for robustness.** Qualitative results with varying random seeds show our model's robustness against stochastic factors. We visualize the results generated with the seed from 0 to 4.

## B.4 ADDITIONAL QUALITATIVE RESULTS

Fig. 17 displays additional qualitative results. Our qualitative results demonstrate that the outputs of our model have superior robustness and 3D-consistency compared to previous methods.

## B.5 ABLATION ON SPARSE DEPTH INJECTOR

We conducted an ablation study regarding the effectiveness of conditional depth maps. Specifically, we remove the sparse depth injector from the diffusion model that receives an optimized embedding

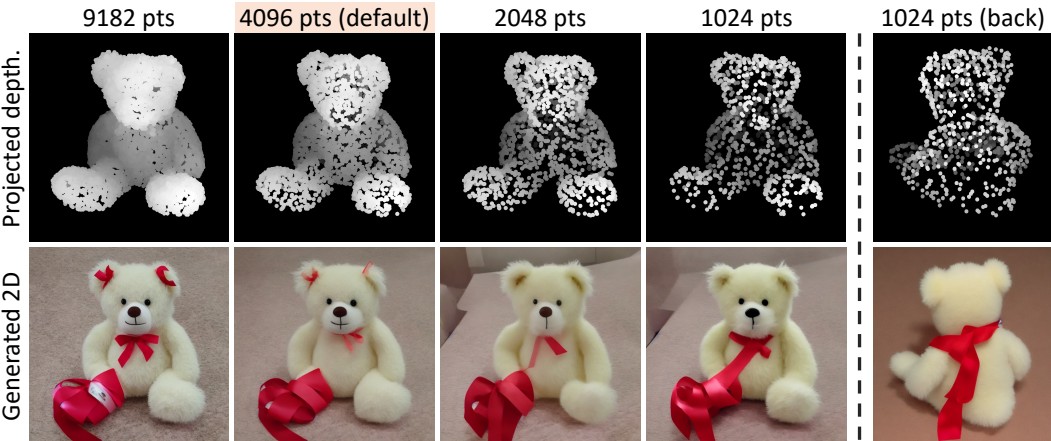

Figure 16: **Sparsity variation with a fixed random seed.**

with LoRA layers. Fig. 18 presents the result of our framework with and without the sparse depth injector, confirming its importance in achieving geometric consistency.

## C  EVALUATION DETAILS

### C.1  USER STUDY

To qualitatively compare our 3DFuse with previous methods, DreamFusion (Poole et al., 2022) and SJC (Wang et al., 2022a), we conduct a user study. As the Imagen (Saharia et al., 2022) model used in DreamFusion is publicly unavailable, we utilize Stable-DreamFusion (Tang, 2022), which uses Stable Diffusion (Rombach et al., 2022). All methods employ Stable Diffusion v1.5 checkpoint.

Our user study questionnaire shows rendered images of 3D scenes generated by 3DFuse, SJC, and Stable-DreamFusion, using identical prompts. We ask the participants to select the result that best fits each of the questions given below. We use 7 prompts and generate three 3D scenes per prompt using 3DFuse, SJC, and Stable-DreamFusion. We give 5 rendered images for each 3D scene, with the camera poses identical across different models for a fair comparison. We do not disclose the models used to generate the results and randomize the order of methods for each question in order to keep the selection process fair. Our final user study is the statistics summarizing responses from a total of 102 participants.

The questions used in the user study are as follows:

- These images are rendered by rotating the camera horizontally from 0 to 360 degrees. Which result has the most natural shape? (3D coherence)
- Which result seems to follow the user input text the best? (prompt adherence)
- Which result has the best overall quality? (overall quality)

### C.2  COLMAP-BASED METRIC

We provide additional details of the COLMAP (Schonberger & Frahm, 2016)-based metric we propose. We sample 100 uniformly spaced cameras whose origin points are on a hemisphere of fixed radius at an identical elevation angle, which is 30 degrees in our setting. All cameras are directed towards the center of the hemisphere, and we render 100 images of the 3D scene from these camera viewpoints. Subsequently, COLMAP predicts the camera pose of each image for point cloud reconstruction. We measure the difference between the predicted camera poses of image pairs that were adjacent at the previous rendering stage.

The variance between the different values is used as a metric for the 3D consistency evaluation. As initial differences between all adjacent camera pairs are identical at the rendering stage, a high variance in predicted difference values indicates the magnitude of inaccuracy in the COLMAP optimization process that predicts cameras. Such inaccuracies are caused by distortions and repeating

Stable-DreamFusion          SJC          **3DFuse (Ours)**

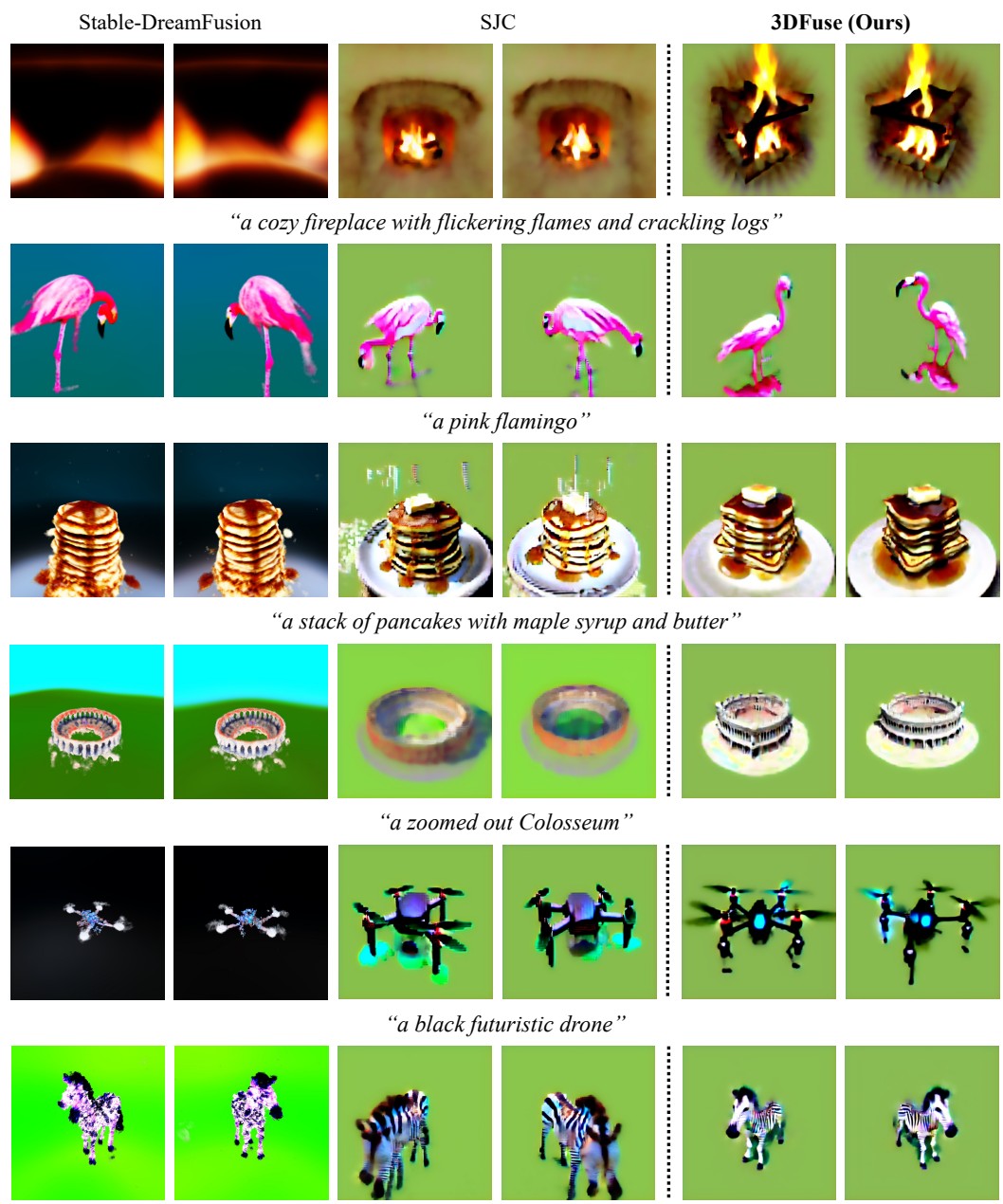

*"a cozy fireplace with flickering flames and crackling logs"*

*"a pink flamingo"*

*"a stack of pancakes with maple syrup and butter"*

*"a zoomed out Colosseum"*

*"a black futuristic drone"*

*"a playful zebra with black and white stripe"*

Figure 17: **More qualitative results.** Additional qualitative results show 3DFuse's superior performance in ensuring geometric and semantic consistency.

artifacts in a 3D scene, whose ambiguous and repetitive nature confuses COLMAP optimization by offering multiple erroneous solutions in the camera pose prediction process. Therefore, it can be argued that this variance value corresponds to the prominence of 3D inconsistent features that make optimization for COLMAP difficult.

Fig. 19 visualizes the point cloud and the camera poses reconstructed through COLMAP optimization. The left columns show COLMAP optimization results of inconsistent 3D scenes generated by SJC (Wang et al., 2022a), and the right columns show COLMAP optimization results of consistent 3D scenes generated by 3DFuse. It is noticeable that the cameras predicted from 3DFuse's scene are more uniformly spaced and stable than those of SJC. In the left-right figure showing a horse

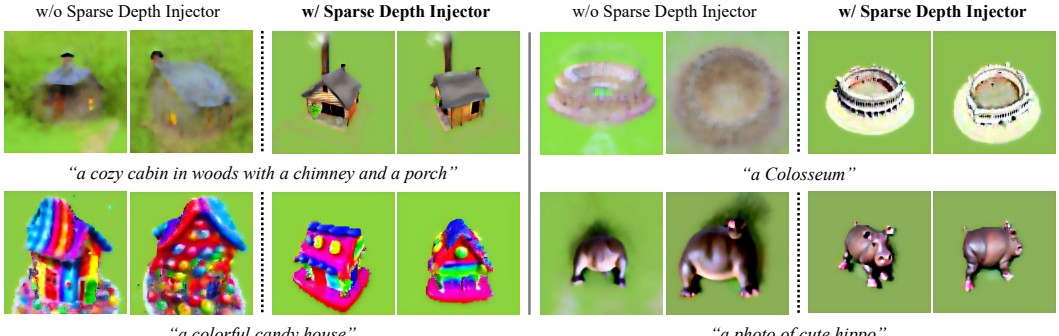

| w/o Sparse Depth Injector | w/ Sparse Depth Injector | w/o Sparse Depth Injector | w/ Sparse Depth Injector |

*"a cozy cabin in woods with a chimney and a porch"*    *"a Colosseum"*

*"a colorful candy house"*    *"a photo of cute hippo"*

Figure 18: **Ablation on sparse depth injector.** Our results demonstrate that the absence of sparse depth injection induces the breakdown of geometry, resulting in blurry and inconsistent 3D shapes, proving the prominence of the sparse depth injector in our framework.

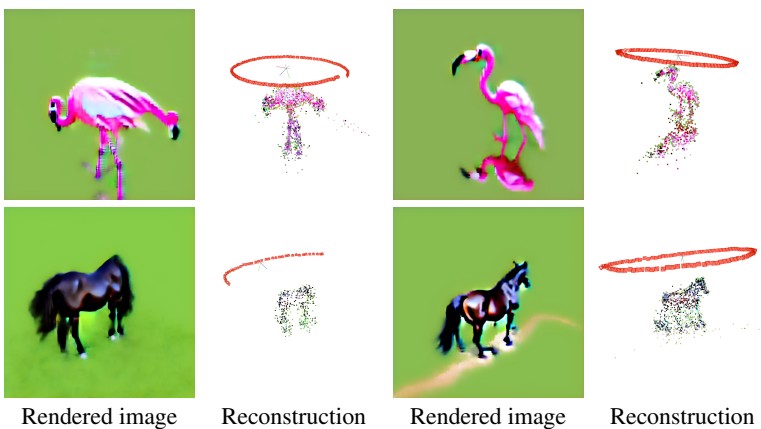

Rendered image    Reconstruction    Rendered image    Reconstruction

Figure 19: **Visualization of predicted camera pose from COLMAP (Schonberger & Frahm, 2016) optimization.** Left columns visualize COLMAP optimization results from the renderings of SJC-generated scenes (Wang et al., 2022a), while the right columns visualize the results from our 3DFuse.

generated by SJC, COLMAP displays evident difficulty in handling the ambiguous, repetitive features of the 3D-inconsistent horse, leading it to mistakenly predict that all cameras are distributed on only one side of the hemisphere. As this phenomenon is consistently observed throughout multiple scenes, we demonstrate that such distortions and 3D inconsistencies increase the difficulty of COLMAP optimization and harm the camera pose prediction quality.

Please note that a limitation of the COLMAP-based metric lies in its resolution. If the generated output lacks sufficient 3D consistency, the optimization process within COLMAP may fail to converge, resulting in significantly lower scores. Consequently, it becomes challenging to differentiate between such samples. However, this metric retains an advantage in assessing the 3D consistency among samples of reasonably high quality.

## C.3    LIMITATION OF CLIP-BASED METRIC

For 2D image generation, metrics such as FID (Heusel et al., 2017) and Inception score (Salimans et al., 2016) are used to quantify the generation quality by measuring the similarity between the distribution of ground truth images and generated images. However, the same approach cannot be applied to score distillation-based text-to-3D generation, due to the absence of 3D scenes corresponding to the text prompts. This presents a challenge for conducting a quantitative evaluation of zero-shot text-to-3D generative models, which previous works have bypassed by showcasing var-

(a) Average of CLIP score : 19.9

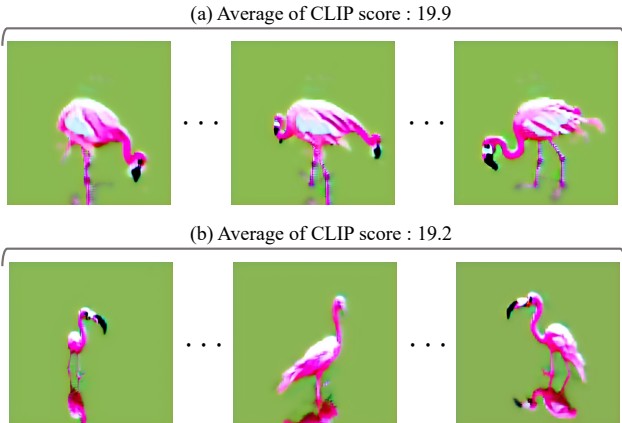

(b) Average of CLIP score : 19.2

Figure 20: **Scenes evaluated by CLIP-based metric.** (a) An inconsistent 3D scene and (b) a consistent 3D scene. The inconsistent 3D scene achieves a higher score in the CLIP-based metric.

ious qualitative results instead of quantitative evaluation (Wang et al., 2022a) or conducting user study (Lin et al., 2022).

Another approach used in previous works (Jain et al., 2022; Poole et al., 2022) is to use CLIP-based metrics, such as CLIP R-precision, which measures retrieval accuracy based on CLIP (Radford et al., 2021) through projected 2D image and text input. Nevertheless, such CLIP-based metrics are ineffective in measuring the quality and consistency of generated 3D geometry. For example, when geometric failure such as a multiple-face problem occurs, producing frontal geometric features multiple times around the 3D scene, the CLIP-based metric measures all such viewpoints with frontal features as having high similarity with the text prompts. This causes such 3D scenes to have misleadingly high scores despite having incorrect geometry. Fig. 20 illustrates an example of this phenomenon, where each image rendered in a geometric inconsistent scene (first row) achieves a higher CLIP score than each image generated in a geometric consistent scene (second row).

## D    LIMITATION

We have proposed a novel 3DFuse framework that infuses 3D awareness into a pretrained 2D diffusion model while preserving its original generalization capability. While 3DFuse stably optimizes NeRF for text-to-3D generation, demonstrating superior performance, there are still several limitations that need to be addressed. First, our approach inherits the difficulty in reflecting complex user prompts faithfully, as it relies on the ability of the pretrained diffusion model to follow text prompts. Additionally, our approach may face various societal biases inherent in the dataset (Schuhmann et al., 2021), similar to the text-to-image generation models (Rombach et al., 2022).

