# OpenReview forum: "Let 2D Diffusion Model Know 3D-Consistency for Robust Text-to-3D Generation"
_ICLR.cc/2024/Conference — ICLR 2024 poster_

### Official Review · Reviewer_s3AW · 2023-10-29

**Soundness:** 3 good
**Presentation:** 3 good
**Contribution:** 3 good
**Rating:** 8
**Confidence:** 5

**Summary:**

The paper investigates the geometry quality of SDS-based methods. By using an additional control net module that is aware of the sparse depth from point-e/mcc, the model grounds the 3D generation with rough geometry and therefore reduces the janus issues. The proposed pipeline can be integrated with SDS and VSD.

**Strengths:**

+ The depth-aware controlnet module alleviates the janus problem and improves the geometry quality of 3D generation greatly.

+ LoRA is utilized for parameter-efficient fine-tuning when improving the semantic consistency.

+ I played with the code of the proposed framework, and it works well.

**Weaknesses:**

- Missing references on image-to-3D generation (CVPR'23 papers, both released on arxiv before the original release of this submission). [1] proposes textual inversion + dreambooth method for improving semantic consistency, similar to Imagic. [2] uses textual inversion for semantic consistency. Would be nice if these approaches were discussed in the related works section, as they are relevant to the semantic consistency component in this work.

[1] Xu D, Jiang Y, Wang P, et al. NeuralLift-360: Lifting an In-the-Wild 2D Photo to a 3D Object With 360deg Views[C]//Proceedings of the IEEE/CVF Conference on Computer Vision and Pattern Recognition. 2023: 4479-4489.

[2] Melas-Kyriazi L, Laina I, Rupprecht C, et al. Realfusion: 360deg reconstruction of any object from a single image[C]//Proceedings of the IEEE/CVF Conference on Computer Vision and Pattern Recognition. 2023: 8446-8455.

**Questions:**

What are the major differences between the proposed method and the above [1][2] on improving semantic consistency in SDS?

---

> ### Author Response · Authors · 2023-11-20
> **Author Response**
>
> Thank you for your constructive review and helpful suggestions! We give a detailed response to your questions and comments below. If any of our responses do not adequately address your concerns, please let us know and we will get back to you as soon as possible.
>
> We also would like to thank the reviewer for the careful review noting the strengths of our paper, as well as verifying our model’s effectiveness by firsthand using its implementation. We would also be thankful if you could check out our **new supplementary video** and our **revised paper**, which contain additional interesting,  high-quality results that are competitive with contemporaneous text-to-3D methods. Thank you for your review and comments.
>
> &nbsp;
>
> ## **W1. Missing references on image-to-3D generation.**
>
> Thank you for your constructive comment. Following your advice, we have added both approaches with discussions in the Related Work section of our revised paper, and we would be thankful if you could view it. We are very grateful for your helpful advice in strengthening our paper.
>
> &nbsp;
>
> ## **Q1. Major differences between the proposed method and previous works on improving semantic consistency in SDS.**
>
> Thank you for pointing this out. We explain the main difference between the semantic consistency used in NeuralLift -360 [a] and RealFusion [b] in two aspects.
>
> ### 1. Regarding differences in the main objective between [a,b] and ours.
>
> Both NeuralLift and Realfusion aim to leverage Stable Diffusion's generative capability to modify text-to-3D generative methodologies into **one-shot 3D reconstruction methods**, where they aim for pixel-level accurate alignment and reconstruction of the input image into a full-fledged 3D model. To this end, they apply direct reconstruction loss as well as score distillation to generate 3D models aligned to accurately input images.
>
> On the other hand, the main objective of our work is fundamentally a **text-to-3D** generation. In our work, we do utilize an image as a stepping stone for point cloud generated and text embedding optimization, but we do not aim for perfect image-to-3D reconstruction like the above papers through these features: instead, our model solely aims for stronger robustness in geometric and semantic consistency of the generated models - for quality enhancement. In this manner, the two works and our 3DFuse tackle a fundamentally different problem in 3D generation.
>
> ### 2. Regarding differences in technical components.
>
> - NeuralLift-360:  This model uses a monocular depth prediction model coupled with pixel-level reconstruction loss from the image to facilitate the generation of **geometry around the image viewpoint**.
> - RealFusion: This method mentions in its Appendix that the incorporation of mono-depth prediction was not effective in facilitating image-to-3D generation: however, this work aims to make the generated scene semantically consistent with a given image through textual inversion, while also using **reconstruction loss to fit the 3D scene** to the image given.
> - 3DFuse (Ours): Our method leverages a generated point cloud to ensure geometric consistency globally across all viewpoints of the scene. While point cloud is coarse and sparse in comparison to mono-depth predictions, this problem is well addressed by our Sparse Depth Injector.  In this process, to ensure semantic consistency we use semantic coding (LoRA + inversion), this time ensuring semantic consistency across all viewpoints of the scene.
>
> We are grateful to the reviewer for your construction question, and we believe this discussion will help strengthen our paper greatly. We will add an additional section in the Appendix to discuss the above comparison in detail. Thank you for your time and effort in reviewing our paper.
>
> &nbsp;
>
> ### Reference
>
> [a] Xu, Dejia, et al. ”NeuralLift-360: Lifting An In-the-wild 2D Photo to A 3D Object with 360° Views”, CVPR 2023
>
> [b] Melas-Kyriazi, Luke, et al. “RealFusion: 360° Reconstruction of Any Object from a Single Image”, CVPR 2023

---

> ### Author Response · Authors · 2023-11-22
> **Follow Up Reminder**
>
> Dear Reviewer s3AW,
>
> Thank you for your time and effort in reading our response! We hope our response has addressed your concerns. If you still feel unclear or concerned, please kindly let us know and we will be more than glad to further clarify and discuss any further concerns. If you feel your concerns have been addressed, please kindly consider if it is possible to update your score.
>
> Thank you!
>
> Paper2347 Authors

---

> ### Author Response · Authors · 2023-11-23
> **Follow Up**
>
> Dear Reviewer s3AW,
>
> We appreciate your time and effort in reading our response and revision! If you still have further concerns or feel unclear, please kindly let us know and we are happy to further clarify and discuss. If you feel your concerns have been addressed, we would appreciate it if you might kindly consider updating the score.
>
> As the discussion deadline is in a few hours, we really look forward to your feedback.
>
> Thank you!
>
> Paper2347 Authors

---

> ### Author Response · Authors · 2023-11-23
> **Thank You**
>
> Dear Reviewer s3AW,
>
> Thank you for raising our score. We would like to thank you again for your constructive review. We sincerely appreciate your suggestions.
>
> Paper2347 Authors

---

### Official Review · Reviewer_MdgG · 2023-11-01

**Soundness:** 3 good
**Presentation:** 3 good
**Contribution:** 2 fair
**Rating:** 5
**Confidence:** 5

**Summary:**

This paper introduces 3DFuse, a new method to integrate 3D consistency information into the SDS loss optimization pipeline of text-to-3D generation. The authors first fine-tune the Stable Diffusion model to understand sparse depth conditions.
During SDS optimization, 3DFuse reconstructs a coarse point cloud from an image using off-the-shell methods and then renders sparse depth maps from the point clouds for depth-conditioned SDS loss supervision.
Both qualitative and quantitative analyses underscore the efficacy of conditioning based on coarse point clouds, resulting in enhancements in the text-to-3D generation results. Additionally, the authors have unveiled a semantic coding method to tackle inherent semantic ambiguities.

**Strengths:**

1. 3DFuse offers a robust solution to fine-tune a sparse depth-conditioned Stable diffusion model, employing the Co3D dataset. The result in Fig.3 provides a compelling argument for the performance of the depth-conditioned diffusion model. Furthermore, depth-conditioned SDS loss augments the 3D consistency of generated results, as depicted in Fig.4.
2. The authors explore the semantic ambiguity problem, which has often been overlooked in prior studies. They introduced semantic code sampling to mitigate this challenge, and Fig.12 convincingly showcases the effectiveness of this strategy.
3. The authors conduct comprehensive ablation studies and extensive experimental validation. Most of the proposed components are well-ablated. In addition, the authors also provide a user study in Fig.8.
4. This paper boasts of a coherent and lucid narrative structure, facilitating easy comprehension.

**Weaknesses:**

1. 3DFuse relies heavily on off-the-shelf point cloud reconstruction or generation methods like Point-E, to obtain coarse point clouds. These point clouds subsequently serve as conditional information in the SDS optimization process. However, sparse-depth renderings are low-quality and may have many artifacts, resulting in degenerated and ambiguous results as shown in Fig.4.
2. The authors provide the novel view synthesis results compared with Zero-123 in Fig.10. However, 3DFuse fails to fit the reference image and generates a different object from it. Furthermore, the predicted novel views are also blurry.
3. The quality of 3DFuse's results lags behind contemporaneous methods. Additionally, it is noteworthy that the results for benchmark methods, such as Dreamfusion and ProlificDreamer, presented in this paper, appear to be less robust than those delineated in their original publications and open-source implementations. It would be beneficial for the authors to recalibrate their methodology based on more formidable baseline models.

**Questions:**

NA

---

> ### Author Response · Authors · 2023-11-20
> **Author Response (1/4)**
>
> Thank you for your constructive review and helpful suggestions! We give a detailed response to your questions and comments below. If any of our responses do not adequately address your concerns, please let us know and we will get back to you as soon as possible.
>
> &nbsp;
>
> ## **W1. Regarding the quality of point cloud obtained from off-the-shelf point cloud models.**
>
> Thank you for pointing this out. We emphasize that one of the main contributions of our paper, **“Sparse Depth Injector (Section 4.3)”,** specifically focuses on addressing the problem of unstable artifacts and errors within the generated 3D point clouds you have mentioned. As we have stated in the *"*Training the sparse depth injector*"* section of our paper, our depth injector **"successfully infers dense and robust structural information without needing any auxiliary network for depth completion"** due to the generative capability of the diffusion model.
>
> We demonstrate this capability in detail in Figure 3 of our paper, where our model equipped with the Sparse Depth Injector implicitly infers the overall geometry from the sparse and erroneous point cloud depth map given and **robustly generates a realistic, artifact-free image in accordance with the general shape of the given sparse depth map.** In comparison, the same sparse point cloud map given to previous depth-conditioning diffusion models [a, b] results in **disfigured, erroneous images**, suffering from the exact problem that you have mentioned.
>
> We find that this ability to infer robust geometric information from sparse point clouds naturally extends to its **robust handling of erroneous point clouds and artifacts** given to it as a condition. Though there may be errors and artifacts in the predicted point cloud, our diffusion model implicitly ignores them in the process of generating realistic images, effectively having an effect of filtering out such point cloud errors when applied for 3D generation. This effect is clearly demonstrated in Figure 11 and Figure 15 of our paper, where the conditioning point cloud of the cup evidently has **erroneously missing regions near the top and the bottom**, but our model generates a whole, robust 3D cup nonetheless (Fig 11), and Figure 15 in our Appendix shows that our model is robust to the sparse nature of the point clouds, generating consistent results despite varying number and sparsity (and therefore increasing ambiguity) of the conditioning point clouds.

---

> ### Author Response · Authors · 2023-11-20
> **Author Response (2/4)**
>
> &nbsp;
>
> ## **W2. Regarding comparison with Zero-123.**
>
>
> ### 1. Our model is fundamentally a text-to-3D generative method and does not attempt to explicitly fit images to 3D scenes.
>
> Thank you for your comment. First, we would like to point out and clarify that our model is **fundamentally a text-to-3D generation method**, and therefore its image-to-3D stage **does not conduct a “strict” image-to-3D modeling (i.e. novel view synthesis)** such as Zero-123 [c] which aims for novel view synthesis and strives to generate pixel-aligned 3D scenes from the image. Instead, in our model, the generated image is used as a ***loose condition*** from which the general structure of the scene (defined by the 3D point cloud) is optimized. Please note that we give no direct loss between the image and the 3D scene for alignment (or fitting) of the scene to the image, and **we do not attempt in our work to achieve such image-to-3D alignment** - nor do we claim that we can do so. For this reason, 3DFuse does not appear to fit the reference image and generates objects not identical to the given input image, as you have mentioned.
>
> However, we point to the fact that even without such image-to-3D losses, generated results show that our 3D scenes generated from the image (shown in Figure 10) **remain semantically consistent along various viewpoints of the scene**. This is due to the **semantic coding**, one of our main contributions, which effectively finetunes the diffusion model and implicitly ensures strong semantic consistency within the 3D scene. As ensuring geometric and semantic consistency across the entire scene is the main motivation of our work, this characteristic aligns nicely with the purpose of our model.
>
> Lastly, we believe your understanding of the latter part of our method as conducting explicit novel view synthesis may have been sparked by our comparative experimentation with the Zero-123 model that does novel view synthesis. We wish to emphasize that our main motivation for this comparison, as we have initially stated in our paper, was that both zero-123 and our work have **similar motivations of giving 3D awareness to existing 2D diffusion models**. It was not our intention to compare the performance of image-to-3D prediction, or novel view synthesis, on the two models, but we may have not been clear enough about this in our paper. We apologize for this possible misunderstanding, and we have revised the writing in this section of our paper to make our point clearer. Thank you for your constructive review in making our paper more robust and clear.
>
> Also, we are grateful for your interpretation of our model as a partial image-to-3D model - which we believe would be a very interesting direction to take our work forward with the incorporation of losses and components that strengthen image-to-3D alignment! We thank you very much for your inspiring comment.
>
> &nbsp;
>
> ### 2. The blurriness of predicted novel views derives from the architecture of Zero-123’s optimization backbone, SJC, and we will soon add results with a different NeRF backbone for non-blurry renderings of the scene.
>
> Thank you for pointing this out. The reason for the blurry novel views that you have mentioned, is due to the architectural choice of the Score Jacobain Chaining (SJC) [d] backbone that the **official Zero 1-to-3 implementation** has used. SJC’s backbone NeRF does not directly model pixel-level RGB but instead models latent 3D features, so **it initially renders a low-resolution feature image and then decodes it into an RGB image through a CNN** (as described in Section 5.2. of the SJC paper), resulting in blurry, aliasing-like effects that you have noticed. Please refer to our response to the second weakness (W2) pointed out by Reviewer ftV7 for additional details.
>
> Following your comment, in our revised paper, we will soon add new qualitative results of Zero-123 in our Appendix, where we remove the blurry effect and acquire clearer results. Specifically, in both Zero-123 and our model, we have replaced feature-level NeRF that SJC has utilized with the Instant-NGP [e] backbone, which directly models and **renders pixel-level RGB,** resulting in clearer, more detailed geometry. We are in the midst of running said experiments (due to resource shortage on our part), and we will add the results to the Appendix as soon as possible.

---

> ### Author Response · Authors · 2023-11-20
> **Author Response (3/4)**
>
> &nbsp;
>
> ## **W3. Regarding 3DFuse’s results in comparison to contemporaneous methods.**
>
> ### 1. With ProlificDreamer backbone, we provide additional results demonstrating our model’s competitiveness to contemporaneous methods.
>
> Thank you very much for your constructive comment. First, we would like to acknowledge that in our initial submission, as the majority of our experimental results utilized as its baseline Score Jacobian Chaining (SJC) [d] - an older methodology showing lower fidelity, its performance in concerning fidelity and detail quality in comparison to contemporaneous works were not clearly displayed.
>
> Therefore, following your advice, we show how our model exhibits **high-fidelity performance competitive to best contemporaneous methods** using the more formidable ProlificDreamer [f] baseline, which is far more high-fidelity and recently introduced than SJC. To more clearly demonstrate how our model achieves significant quality and fidelity improvement over its baseline, we provide **extensive additional results** in our **new supplementary video** and also in Figure 4 of our revised paper. Our model is shown to handle **complex prompts** with geometric robustness, while the same prompts induce failures at the baseline model.
>
> This demonstrates that our model can achieve **diverse, highly detailed results with robust geometry** when attached to more formidable baselines. This lets us to conjecture that if more powerful SDS-based text-to-3D models are introduced in the future, our model will likewise be very effective in enhancing its performance both in terms of geometry and fidelity.
>
> &nbsp;
>
> ### 2. We provide additional results showing that the robustness and performance of our unofficially implemented ProlificDreamer baseline is similar to the official ProlificDreamer results.
>
> Thank you for your comment. You have commented that the results for the benchmark methods (such as Dreamfusion and ProlificDreamer) appear to be less robust than those delineated in their original publications and open-source implementations. As the official implementation for both Dreamfusion and ProlificDreamer **has not yet been officially revealed**, we have instead used unofficial Threestudio open-source implementations of said models as our baseline models.
>
> However, we argue that the Threestudio-version models do not lack performance or robustness compared to those delineated in original publications. We observe that when we generate 3D models with our unofficial baseline **using the same text prompts** as those given in their original publications and project page, its output shows **similar robustness and quality** to those shown in the original publications. We demonstrate this in Part 3 of our **supplementary video**, where we compare videos of 3D models curated and presented on the official ProlificDreamer project page and those generated by our baseline, and it can be clearly seen that the results’ quality is very similar.
>
> Therefore, we speculate that the lack of robustness of the results that you have mentioned does not come from the lack of robustness of the baselines we have used, but the models’ **fundamental weakness** to complex prompts - especially prompts referring to objects with an explicit frontal ***face*** - that are likely to induce geometric inconsistency problems, which is likely to not be as clearly delineated in the original publications due to their broken geometry and quality. As our main motivation is addressing this issue of geometry inconsistencies, we have delineated such broken results more clearly to demonstrate our method’s effectiveness in solving these issues, and our method is **very capable in this aspect** as Reviewer s3AW has verified.

---

> ### Author Response · Authors · 2023-11-20
> **Author Response (4/4)**
>
> ### 3. Our model adds powerful controllability to existing baselines.
>
> Lastly, we also call to attention another important advantage of our model, which is how our model **adds significant controllability** to existing text-to-3D baselines. As we have described in Section 4.3 of our paper, “our method enables the overall shape of the scene to be decided before the lengthy optimization process”, allowing the user to more **meticulously control the generation process** and generate specific 3D scenes tailored to their needs.
>
> To demonstrate this controllability aspect of our method, we include additional experimental results in Part 2 of the **new supplementary video**, demonstrating how **incredibly diverse results under the same text prompt** can be generated under the **user’s control** due to our method’s unique point cloud conditioning.
>
> This clearly shows that even in cases of text prompts where the baseline model and our method both display exceptional results (thus ours showing relatively marginal improvement compared to baseline in such cases), our method still holds a **strong advantage over the baseline** regarding the **controllability** of generated scenes, allowing the user to generate incredibly diverse results as one wishes. We would be grateful if you could take this fact into consideration as well. Thank you for your time and effort in reviewing our paper.
>
> &nbsp;
>
> ### Reference
>
> [a] Rombach, Robin, et al. “High-resolution image synthesis with latent diffusion models”, CVPR 2022
>
> [b] Zhang, Lvmin, et al. “Adding conditional control to text-to-image diffusion models”, ICCV 2023
>
> [c] Liu, Ruoshi, et al. “Zero-1-to-3: Zero-shot one image to 3d object”, ICCV 2023
>
> [d] Wang, Haochen, et al. “Score jacobian chaining: Lifting pretrained 2d diffusion models for 3d generation” CVPR 2023
>
> [e] Müller, Thomas, et al. “Instant neural graphics primitives with a multiresolution hash encoding”, SIGGRAPH 2022
>
> [f] Wang, Zhengyi, et al. “ProlificDreamer: High-Fidelity and Diverse Text-to-3D Generation with Variational Score Distillation”, NeurIPS 2023

---

> ### Author Response · Authors · 2023-11-21
> **Additional Author Response**
>
> &nbsp;
>
> ## Additional experiment results regarding image-to-3D models
>
> Following your comment, we have updated the Appendix of our **revised paper with additional experiments** concerning our comparison to Zero-123, and to another image-to-3D method that is concurrent, Magic-123 [a], following the comment of Reviewer ftV7. As we have previously explained in our response, a direct comparison of our model to existing image-to-3D works is difficult, as our model is fundamentally a text-to-3D model, and does not attempt pixel-level fitting of the image to the 3D scene. Backgrounds of the generated images are masked out using preprocessing methods that each paper has originally used. Therefore, to compare our model to these models in an equal text-to-3D setting, we first generate images from a single text prompt (through the Stable Diffusion model) and give these images Zero-123 and Magic-123 for 3D scene generation.
> Our 3DFuse undergoes its original generation process with the identical text prompt, and the images given to each model are not curated. Also, to remove the **aliasing-like effect in the renderings** you have mentioned, we have changed the backbone NeRF model of Zero-123 and 3DFuse from feature-level SJC to **pixel-level Instant-NGP backbone**, allowing us to gain clearer results.
>
> The result is given in Figure 14 of our revised Appendix, and the generated results no longer exhibit the aliasing effect shown previously in our comparison to Zero-123, verifying our comment that attributes said issue to NeRF backbone architecture. The experiment results show that our method’s generative capabilities enable our model to generate more detailed, high-fidelity results in comparison to image-to-3D models that attempt explicit novel view synthesis of conditioning images. We hypothesize the reason for this is that strict image-to-3D models struggle to predict novel views of an image when an image is outside the domain of training data, while our model is loosely conditioned and therefore gives the generative model more freedom to generate realistic details, fully realizing its generative capability. This is in agreement with the analysis that we have given in our comparison to Zero-123 at our main paper.
>
> &nbsp;
>
> ## Reference
>
> [a]  Qian, Guicheng, et al. “Magic123: One Image to High-Quality 3D Object Generation Using Both 2D and 3D Diffusion Priors”, ArXiv preprint

---

> ### Author Response · Authors · 2023-11-22
> **Follow Up Reminder**
>
> Dear Reviewer MdgG,
>
> Thank you for your time and effort in reading our response! We hope our response has addressed your concerns. If you still feel unclear or concerned, please kindly let us know and we will be more than glad to further clarify and discuss any further concerns. If you feel your concerns have been addressed, please kindly consider if it is possible to update your score.
>
> Thank you!
>
> Paper2347 Authors

---

> ### Author Response · Authors · 2023-11-23
> **Follow Up**
>
> Dear Reviewer MdgG,
>
> We appreciate your time and effort in reading our response and revision! If you still have further concerns or feel unclear, please kindly let us know and we are happy to further clarify and discuss. If you feel your concerns have been addressed, we would appreciate it if you might kindly consider updating the score.
>
> As the discussion deadline is in a few hours, we really look forward to your feedback.
>
> Thank you!
>
> Paper2347 Authors

---

### Official Review · Reviewer_ftV7 · 2023-11-07

**Soundness:** 4 excellent
**Presentation:** 3 good
**Contribution:** 4 excellent
**Rating:** 6
**Confidence:** 4

**Summary:**

This work introduces a new framework called 3DFuse that enhances the robustness and 3D consistency of score distillation-based methods for text-to-3D generation. The framework incorporates 3D awareness into the pretrained 2D diffusion model, resulting in geometrically consistent and coherent 3D scenes. The authors also introduce a new technique called semantic coding for improved results. The effectiveness of the framework is demonstrated through qualitative analyses and ablation studies, and it is shown to outperform previous prompt-based methods in terms of precision and controllability of injected 3D awareness. The 3DFuse framework and semantic coding technique have the potential to improve the quality and controllability of generated 3D scenes, which could have applications in various fields such as virtual reality, gaming, and architecture.

**Strengths:**

- The authors introduce a novel framework called 3DFuse that enhances the robustness and 3D consistency of score distillation-based methods for text-to-3D generation. This framework incorporates 3D awareness into the pretrained 2D diffusion model, resulting in geometrically consistent and coherent 3D scenes.

- The authors introduce a new technique called semantic coding that involves generating an initial image based on the text prompt and optimizing the corresponding text prompt embedding. This technique improves the quality and controllability of generated 3D scenes.

- The authors demonstrate the effectiveness of the 3DFuse framework through qualitative analyses, which show that it outperforms previous prompt-based methods in terms of precision and controllability of injected 3D awareness.

- The 3DFuse framework and semantic coding technique have the potential to improve the quality and controllability of generated 3D scenes

**Weaknesses:**

- My biggest concern is the marginal improvement. It seems that the proposed model only shows a very limited improvement compared to each baseline method.

- My understanding is that the video from supp is generated by render well-optimized NeRF, which should be inherently. But why does the video itself look aliasing?

- The proposed framework breaks the text-to-3D problem into text-to-image + image-to-3D tasks. Therefore, it would be better to compare it with other image-to-3D methods [a, b] too.

References:

[a] Zero-1-to-3: Zero-shot One Image to 3D Object

[b] Magic123: One Image to High-Quality 3D Object Generation Using Both 2D and 3D Diffusion Priors

**Questions:**

- How stable is the 3D point cloud generative model? Any failure cases of predicted 3D point cloud? How to handle it?

- Why should it predict a point cloud and then convert it to a depth map? What if the proposed module directly predicts a monocular depth map?

---

> ### Author Response · Authors · 2023-11-20
> **Author Response (1/3)**
>
> Thank you for your constructive review and helpful suggestions! We give a detailed response to your questions and comments below. If any of our responses do not adequately address your concerns, please let us know and we will get back to you as soon as possible.
>
> &nbsp;
>
> ## **W1. Regarding marginal improvement over the baseline method.**
>
> Thank you for your comment. First, we respectfully would like to ask for clarification: could you explain to us in further detail in which aspect you have deemed our model to show ‘marginal improvement’? We understand you mean fidelity improvement which pertains to the general detail quality and realism of generated 3D scenes, and we provide our three-part response to this weakness under such an assumption. If we have misunderstood, please tell us and we will provide additional responses accordingly.
>
> &nbsp;
>
> ### 1. Our method is a plug-and-play module specifically designed for resolving geometric inconsistency problems of baseline text-to-3D methods.
>
> We first emphasize that the main focus of our methodology was designing a **plug-and-play module** that is capable of **resolving the geometric inconsistency problem** of whichever score distillation sampling (SDS)-based text-to-3D baseline it is attached to. Therefore, while our model greatly enhances the geometric consistency of each baseline, as clearly demonstrated by Figures 4 and 5, its characteristics (such as fidelity of results) tend to follow those of the baseline models and their architecture.
>
> &nbsp;
>
> ### 2. Our model achieves drastic quality improvement, and we additionally provide extensive results to demonstrate it.
>
> Nevertheless, we find that such capturing of geometric consistency indeed **facilitates convergence, naturally leading to increased fidelity and quality** of our results in comparison to the baseline. However, in hindsight, we understand and agree with your comment that the effect was not clearly visible in our paper because the majority of our experimental results utilized Score Jacobian Chaining (SJC) [a], an older methodology showing lower fidelity, as its baseline.
>
> Therefore, to more clearly demonstrate how our model achieves significant quality and fidelity improvement over its baseline, we provide **additional results** in Figure 4 of our **revised paper,** and also in our **new supplementary video.** We show how our model exhibits outstanding performance using the ProlificDreamer [b] baseline, which is far more high-fidelity and recently introduced than SJC. Our model is shown to handle highly **complex prompts** with geometric robustness, while the same prompts induce failures at the baseline model. Through these results, we demonstrate that our model achieves **diverse, highly detailed results with robust geometry** when attached to more powerful baselines.
>
> This enables us to conjecture that if more powerful SDS-based text-to-3D models are introduced in the future, our model will likewise be very effective in enhancing its performance both in terms of geometry and fidelity. We will provide more extensive results in the Appendix to showcase our model’s capability.
>
> &nbsp;
>
> ### 3. Our model adds powerful controllability to existing baselines.
>
> Lastly, we also call to attention another important advantage of our model, which is how our model **adds significant controllability** to existing text-to-3D baselines. As we have described in Section 4.3 of our paper, “our method enables the overall shape of the scene to be decided before the lengthy optimization process”, allowing the user to more **meticulously control the generation process** and generate specific 3D scenes tailored to their needs.
>
> To demonstrate this controllability aspect of our method, we include additional experimental results in Part 2 of our **new supplementary video**, demonstrating that **incredibly diverse results under the same text prompt** can be generated under the **user’s control** due to our method’s unique point cloud conditioning. We will also include these results in our revised paper.
>
> This clearly shows that even in cases of text prompts where the baseline model and our method both display exceptional results (thus ours showing **marginal improvement** compared to baseline in such cases), our method still holds a **strong advantage over the baseline** regarding the **controllability** of generated scenes, allowing the user to generate incredibly diverse results as one wishes. We would be grateful if you could take this fact into consideration as well.

---

> ### Author Response · Authors · 2023-11-20
> **Author Response (2/3)**
>
> ## **W2. Regarding the aliasing-like effect in the supplementary video.**
>
> The reason for the aliasing-like effect that you have mentioned, shown in some of the renderings in our supplementary video, is due to the architectural choice of the SJC backbone that we have used. SJC’s backbone NeRF does not directly model pixel-level RGB but instead models latent 3D features, so **it initially renders a low-resolution feature image and then decodes it into an RGB image through a CNN** (as described in Section 5.2. of the SJC paper), resulting in aliasing-like effects that you have noticed. As this is an issue that stems from the original SJC model that its authors have uploaded, notice that the results from the baseline model also evidently display such effects.
>
> Due to this reason, our experimental results from the ProlificDreamer baseline, which models and **renders pixel-level RGB directly with Instant-NGP [c], do not display such an aliasing-like effect.** We provide additional results from the ProlificDreamer baseline in both our revised paper and the new supplementary video **along with their normal maps** for a clearer demonstration of how the aliasing effect is removed, so please take a look into it. This clearly shows that **this issue does not derive from any component of our proposed methodology, but rather comes from the architecture of the NeRF backbone of the baseline model that we use.**
>
>
> &nbsp;
>
> ## **W3. Regarding comparison to image-to-3D methods.**
>
> Thank you for your constructive comment. We wholly agree with your advice that providing comparisons on the image-to-3D tasks will definitely help strengthen our paper, and will update our paper accordingly.
>
> ### 1. We kindly refer to our comparison with Zero-1-to-3 [d].
>
> In this light, we kindly refer to the "Comparison with Zero-1-to-3" in Section 5.4 (Ablation Study) of our paper, in which we had already given a detailed comparison and discussion comparing our work to Zero-1-to-3 in our initial submission.
>
> ### 2. Clarification regarding the image-to-3D component of our model.
>
> We would also like to point out that our model is **fundamentally a text-to-3D generation method**, and therefore its image-to-3D stage does not conduct a “strict” image-to-3D modeling such as Zero-123, which aims to generate pixel-aligned 3D scenes from the image. In our method, **the generated image is used as a loose condition** from which the general structure of the scene (defined by the 3D point cloud) is generated, and we give no direct loss between the image and the 3D scene to fit the scene to the image, **making a strict comparison of our model to an explicit image-to-3D model difficult**. However, our semantic encoding finetunes the diffusion model to the conditioning image, strengthening semantic and geometric consistency across various viewpoints of the 3D scene.
>
> ### 3. We will add our comparison with Magic-123 and Zero-123 in our revised paper.
>
> Also following your advice, we will update our Appendix to include an **additional qualitative comparison** of our work to Zero-123 and Magic-123. We have changed the backbone NeRF model from feature-level SJC to pixel-level Instant-NGP backbone, to remove the “blurry effect” (same as the aliasing effect you have mentioned in W2), deriving clearer results. We are just starting to run said experiments (due to resource shortage on our part), and we will add the results to the Appendix, as soon as possible.

---

> ### Author Response · Authors · 2023-11-20
> **Author Response (3/3)**
>
> ## **Q1. Regarding the stability and failure of the 3D point cloud generative model.**
>
> Thank you for pointing this out. We first would like to mention that we observe that the 3D point cloud generative model we currently use, Point-E [f], shows high stability in image-to-point cloud generation when generating coarse geometry.  As our model leverages these models solely for such coarse geometry, its results seem sufficient in our framework. However, as you have mentioned, there are failure cases where artifacts and errors appear during the point cloud generation process.
>
> In this light, we emphasize that one of the main contributions of our paper, Sparse Depth Injector (described in Section 4.3), specifically focuses on addressing this problem. As we have stated in the "Training the sparse depth injector" section of our paper, our depth injector **"successfully infers dense and robust structural information without needing any auxiliary network for depth completion"** due to the generative capability of the diffusion model.
>
> We find that this ability to infer robust geometric information from sparse point clouds naturally extends to its **robust handling of erroneous point clouds** and artifacts given to it as a condition. Therefore, though there may be errors and artifacts in the predicted point cloud, our diffusion model implicitly ignores them in the process of generating realistic images, effectively having an effect of filtering out such point cloud errors when applied for 3D generation.
>
> This effect is clearly demonstrated in multiple locations throughout our paper. In **Figure 3** of our paper, our model implicitly infers the overall geometry from the sparse and erroneous point cloud depth map given and robustly generates a realistic, artifact-free image in accordance with the general shape of the given sparse depth map. In **Figure 11** of our paper, the conditioning point cloud of the cup evidently has erroneously missing regions near the top and the bottom, but our model generates a whole, robust cup nonetheless.
>
> &nbsp;
>
> ## **Q2. Regarding directly predicting monocular depth map.**
>
> Thank you for your suggestion. The main difficulty in directly using a depth map in our model derives from the fact that depth prediction models **only reconstruct a single side of the given scene,** and are incapable of modeling the 3D geometry of unseen, occluded areas. As it cannot predict the entire shape of the scene but gives us only **incomplete partial geometry**, it does not fit with our main purpose of removing geometric inconsistency issues from **all viewpoints** of the 3D scene, as these issues (such as multi-face Janus problem) mostly occur at side and back regions occluded from the image. Therefore, in order to model the entire scene in 3D, including the side and back regions, we need to leverage generative models that can infer geometries in unseen occluded regions with its generative prior - which is why we mainly resort to sparse point cloud generative model, Point-E.
>
> &nbsp;
>
> ### Reference
>
> [a] Wang, Haochen, et al. “Score jacobian chaining: Lifting pretrained 2d diffusion models for 3d generation” CVPR 2023
>
> [b] Wang, Zhengyi, et al. “ProlificDreamer: High-Fidelity and Diverse Text-to-3D Generation with Variational Score Distillation”, NeurIPS 2023
>
> [c] Müller, Thomas, et al. “Instant neural graphics primitives with a multiresolution hash encoding”, SIGGRAPH 2022
>
> [d] Liu, Ruoshi, et al. “Zero-1-to-3: Zero-shot one image to 3d object”, ICCV 2023
>
> [e] Qian, Guicheng, et al. “Magic123: One Image to High-Quality 3D Object Generation Using Both 2D and 3D Diffusion Priors”, ArXiv preprint
>
> [f] Nichol, Alex, et al. “Point-E: A System for Generating 3D Point Clouds from Complex Prompts”, ArXiv preprint

---

> ### Author Response · Authors · 2023-11-21
> **Additional Author Response**
>
> &nbsp;
>
> ## Additional experiment results regarding image-to-3D models
>
> We have updated the Appendix of our **revised paper with additional experiments** concerning our comparison to image-to-3D models that you have mentioned, Zero-123 and Magic-123. As we have previously explained in our response, a direct comparison of our model to existing image-to-3D works is difficult, as our model is fundamentally a text-to-3D model, and does not attempt pixel-level fitting of the image to the 3D scene.
>
> Therefore, to compare our model to these models in an equal text-to-3D setting, we first generate images from a single text prompt (through the Stable Diffusion model) and give these images Zero-123 and Magic-123 for 3D scene generation. Backgrounds of the generated images are masked out using preprocessing methods that each paper has originally used. Our 3DFuse undergoes its original generation process with the identical text prompt, and the images given to each model are not curated. Also, to remove the **aliasing-like effect in the renderings** you have mentioned, we have changed the backbone NeRF model of Zero-123 and 3DFuse from feature-level SJC to **pixel-level Instant-NGP backbone**, allowing us to gain clearer results.
>
> The result is given in Figure 14 of our revised Appendix, and the generated results no longer exhibit the blurry effect shown previously in our comparison to Zero-123. The 3D scene generation results show that our method’s generative capabilities enable our model to generate more detailed, high-fidelity results in comparison to image-to-3D models that attempt explicit novel view synthesis of conditioning images. We hypothesize the reason for this is that strict image-to-3D models struggle to predict novel views of an image when an image is outside the domain of training data, while our model is loosely conditioned and therefore gives the generative model more freedom to generate realistic details, fully realizing its generative capability. This is in agreement with the analysis that we have given in our comparison to Zero-123 in our main paper.

---

> ### Author Response · Authors · 2023-11-22
> **Follow Up Reminder**
>
> Dear Reviewer ftV7,
>
> Thank you for your time and effort in reading our response! We hope our response has addressed your concerns. If you still feel unclear or concerned, please kindly let us know and we will be more than glad to further clarify and discuss any further concerns. If you feel your concerns have been addressed, please kindly consider if it is possible to update your score.
>
> Thank you!
>
> Paper2347 Authors

---

> ### Author Response · Authors · 2023-11-23
> **Follow Up**
>
> Dear Reviewer ftV7,
>
> We appreciate your time and effort in reading our response and revision! If you still have further concerns or feel unclear, please kindly let us know and we are happy to further clarify and discuss. If you feel your concerns have been addressed, we would appreciate it if you might kindly consider updating the score.
>
> As the discussion deadline is in a few hours, we really look forward to your feedback.
>
> Thank you!
>
> Paper2347 Authors

---

### Author Response · Authors · 2023-11-20
**General Response**

# General Response

We would like to first thank the reviewers for the helpful suggestions and constructive reviews. We are greatly encouraged by their assessment of our work as novel (ftV7), offering a robust (MdgG) solution with compelling argument (MdgG), coherent and well-written (MdgG), effective (ftV7) in greatly improving the generation quality (s3AW, ftV7) and controllability (ftV7). They acknowledge how our paper tackles a problem previously overlooked in prior studies (MdgG) and convincingly showcases the effectiveness of our strategy (MdgG) through qualitative analyses (ftV7), extensive experimental validation (MdgG) and well-ablated comprehensive ablation studies (MdgG), outperforming previous prompt-based methods in precision and controllability (ftV7). We also thank the reviewers for personally verifying our model’s effectiveness (s3AW).

**We provide extensive additional high-fidelity results in our new supplementary video and revised paper.** The main concern that multiple reviewers raised was how our model’s full performance and quality-wise competitiveness to contemporaneous methods were not clearly displayed in the paper. To address this problem, we provide a **new, additional supplementary video** and additional results in our **revised paper** to clearly demonstrate how our model is capable of generating from complex text prompts **high-fidelity, geometrically robust results competitive to contemporaneous methods.**

In the new supplementary video, we also include additional experimental results showing in further detail how our model adds controllability to existing baseline models. We also provide an experiment comparing our baseline’s (unofficial implementation) robustness against the results given in the official paper and project pages to further validate our experimental results. These results will be added to our revised paper quickly as well, with the explanations we provide below.

We would be truly grateful if you could view the video and our revised paper, and take it into consideration. If there are any comments that we did not adequately address despite the revision, they will be thoroughly reflected in the final version of our paper. Thank you for your helpful and constructive reviews.

---

### Meta-Review · Area_Chair_dWHT · 2023-12-08

**Metareview:**

This paper studies the geometric quality of methods based on Sparse Depth Sampling (SDS). By incorporating a depth-aware control net module that utilizes sparse depth data from point-e/mcc, the model establishes a basic geometric framework for 3D generation, thus mitigating the Janus issue. This proposed pipeline can be seamlessly integrated with existing SDS and VSD systems.

Key strengths include:

1. The depth-aware control net module effectively addresses the Janus problem and significantly enhances the geometric quality of 3D generation.

2. The use of LoRA's for parameter-efficient fine-tuning enhances semantic consistency.

3. Practical application of this concept has shown promising results with several SDS based method.

After considering the reviewers' comments, rebuttals, and follow-up discussions, it appears that the authors have adequately addressed most of the primary concerns. The idea of adding an extra 3D-aware conditional channel is a robust and straightforward approach that enhances any SDS-based 3D generation system. This paper thoroughly explores this concept and proposes an effective solution, addressing most raised questions.

Specific concerns, such as image alignment issues in comparison to Z-123, which is image-conditional, are less relevant since this method is designed for text-to-3D conversion. Criticisms regarding the use of off-the-shelf methods for point cloud generation are not as critical in my opinion, as current literature offers several reliable methods for predicting dense depth maps from monocular images.

Another issue raised concerned the marginal improvements over baseline methods. However, I believe the insight and contribution of this paper are both simple and powerful. This approach can be integrated into numerous existing SDS-based methods. The simplicity and universality of this method are its strengths, not weaknesses

For these reasons, I recommend accepting this paper.

**Justification For Why Not Higher Score:**

Evaluating text-to-3D systems at scale has proven challenging, particularly given the rapid increase in the number of papers and approaches in recent months. The paper could have been strengthened with more robust evaluations, but this issue is a general limitation in this area of research, rather than being specific to this particular paper.

**Justification For Why Not Lower Score:**

All questions raised during the review process were adequately addressed.

---

### Decision · Program_Chairs · 2024-01-16

Accept (poster)